# Path Integral Approach to Nondispersive Optical Fiber Communication Channel

**DOI:** 10.3390/e22060607

**Published:** 2020-05-29

**Authors:** Aleksei V. Reznichenko, Ivan S. Terekhov

**Affiliations:** 1Theoretical Department, Budker Institute of Nuclear Physics of Siberian Branch Russian Academy of Sciences, 630090 Novosibirsk, Russia; 2Department of Physics, Novosibirsk State University, 630090 Novosibirsk, Russia

**Keywords:** nonlinear optical fiber channel with zero dispersion, channel capacity, path integral formalism

## Abstract

In the present paper we summarize the methods and results of calculations for the theoretical informational quantities obtained in our works for the nondispersive optical fiber channel. We considered two models: the per-sample model and the model where the input signal depends on time. For these models we found the approach for the calculation of the mutual information exactly in the nonlinearity parameter but for the large signal-to-noise power ratio. Using this approach for the per-sample model we found the lower bound of the channel capacity in the intermediate power range.

## 1. Introduction

For a linear transmission system, Shannon [1] obtained the famous result for the channel capacity that is the maximal amount of information which can be transmitted through the channel with additive noise:(1)$$\begin{array}{c} C\propto \log_{2}\left(1+P/P_{noise}\right)\;, \end{array}$$
where *P* is the input signal power, and $P_{noise}$ is the noise power. In the recent 25 years, the power and frequency bandwidth of the signals transmitted through optical fiber channels have been grown. It results in the necessity to take into account the Kerr nonlinearity when considering the modern optical fiber channels. The Kerr nonlinearity leads to the distortion of the signal and to the nonlinear interaction of the signal with the noise in the information channel. Therefore, for such channels, Shannon’s result (Equation 1) should be modified to take into account the nonlinearity effects.

It is worth noting that nonlinear effects in a channel depend on both a particular realization of the communication channel and physical fiber parameters. When designing the real transmission systems, say, in the wavelength division multiplexing systems, the following nonlinear effects should be taken into account: self-phase modulation, cross-phase modulation and so on [2]. These effects are due to the optical Kerr effect, i.e., the change of the refractive index of fiber material in response to the applied electric field. Concerning other fiber parameters, the second dispersion coefficient is the fundamental one, and the second dispersion coefficient varies from non-zero values (typical value is about $\beta =2\times 10^{-23}\;s^{2}/\mathrm{km}$) to almost zero ones. Around the world, the vast majority of fiber networks are fiberoptic communication channels with non-zero dispersion. However, the design of some fiber channels is arranged in the zero-dispersion region of the wavelength in the transmission windows: in particular, the second transmission window (1310 nm for the single-mode optical fiber) has almost zero dispersion. In the presence of dispersion, the Kerr nonlinearity effect manifests as a phase shift of the signal over distances depending on dispersion value. For large dispersion, these distances are much less than the usual propagation distance. For the nondispersive channel, this phase shift becomes the global effect extended over all propagation distances. The third essential component of a communication channel is signal attenuation. The fiber loss is commonly compensated by (equally) spaced amplifiers. These amplifiers bring in the stochastic noise to the signal in the propagation process [3]. Therefore, the relevant models describing the signal propagation in a communication channel should deal with the following phenomena: Kerr nonlinearity, dispersion, and noise effects. The simplest channel model taking into account all these effects includes the nonlinear Schrödinger equation (NLSE) with additive Gaussian noise [3,4,5].

The model describing NLSE with zero noise is solvable in the formalism of the inverse scattering problem [6,7], referred to as the nonlinear Fourier transformation method for some time past. However, the consideration of NLSE with the additive noise and with non-zero dispersion turned out to be a very difficult problem, since, in addition to effects associated with the nonlinearity, there are some effects related to the dispersion and with the nonlinear interaction of the signal with the noise. There are some papers where both effects of nonlinearity and dispersion are attempted to take into account for a noisy channel, see, e.g., Refs. [8,9,10,11,12,13] and references therein. However, the problem of the channel capacity calculation, e.g., the problem to find the maximal information transmission rate over a given bandwidth is far from the solution.

In the present paper, we concentrate our attention on nondispersive fiber-optic communication channels. Of course, the nondispersive model is in a certain sense a toy model, but it takes into account the effects associated with the Kerr nonlinearity and the noise in the channel. As a consequence, the model includes the effects of the nonlinear interaction of the signal with the noise. Therefore, the model can be used for the rough estimations of the capacity for the fiber-optic communication channels with non-zero dispersion. Besides, the methods developed for the study of the nondispersive fiber-optic communication channels can help for the investigation of realistic nonlinear communication channels with non-zero dispersion.

Considering the fiber-optic channel it is important to clearly define the model of the channel, i.e., it is necessary to define the model of signal propagation, models of the transmitter, receiver, and the model of signal processing (i.e., the recovering of the information from the received signal). The model of propagation should include: the attenuation of the signal in propagation through the optical fiber, the model of amplifiers which compensate the attenuation and add the noise in the channel. The models of the the input signal, receiver, and signal processing reveal itself in the appearance in the channel model such parameters as its frequency bandwidths [14]. In one type of models, the propagation of the signal obeys the NLSE with zero dispersion and with the additional term describing the attenuation compensated by the periodically located amplifiers [15,16]. Another type of the models is considered in Refs. [4,17,18,19,20,21,22], where it is assumed that the amplifiers are continuously distributed in such a way that there is no attenuation, but the amplifiers induce the stochastic noise in the channel. Below we consider this type of model.

In Refs. [4,17,18,19,20,21,22,23] the propagation of the signal is described by the NLSE with zero dispersion, therefore, the signal propagation at different time moments evolves independently. So, formally, one can consider the signal in the fixed time moment, and the signal ceases to depend on time. The model where the signal does not depend on time is referred to as the per-sample model. For the per-sample model in Ref. [4] the conditional probability density function (PDF) was found. In Ref. [17] the authors also found the conditional PDF, and showed that at large signal power the channel capacity obeys the condition
(2)$$\begin{array}{c} C\geq \frac{1}{2}\log_{2}\left(P/P_{noise}\right)+O\left(1\right). \end{array}$$

In Ref. [18] the conditional probability density function also was found, and we attempted to find the lower bound of the per-sample channel capacity for the not-too-large powers of the signal. In Refs. [19,20] the new method of calculation of the conditional probability density function was introduced. The method is based on the calculation of the path-integral using the approach similar to the semi-classical approximation in quantum mechanics. The new method allows us to calculate the statistical properties of the signal such as the entropy of the output signal, the conditional entropy, and the mutual information. All calculations are performed exactly in the nonlinear parameter, but for the large signal-to-noise power ratio (SNR). Using the method we find the probability density function of the input signal that maximizes the mutual information obtained in the leading order in the parameter 1/SNR in the intermediate power range. In the recent paper [22] the important result was obtained: the authors found the upper bound of the capacity for the per-sample model for arbitrary signal powers. The results of Refs. [19,20,22] are consistent.

It is obvious that the per-sample model does not describe the spectral broadening that plays an important role in the capacity limitation. In Ref. [14] it was demonstrated that there is a limit of the input signal power *P* beyond which the fixed bandwidth of the distributed optical amplification does not exceed the bandwidth of the propagating signal (that grows when increasing *P* due to the signal-noise mixing), and therefore, the per-sample model is not relevant anymore. The next reason for the per-sample model limitation is that the per-sample receiver has infinite bandwidth while for the real communication system the receiver bandwidth is limited. Moreover, in Ref. [14] Kramer demonstrated how to achieve the infinite capacity for any power *P* in the per-sample model of Refs. [4,17,18], if one assumes the noise bandwidth to be limited and sends the input signal energy in the noise-free spectrum. In Ref. [21], we applied the methods developed for the per-sample model in Refs. [19,20] to describe nondispersive communication channels with the nontrivial time dependence of the input signal, with a realistic model of the receiver and with a noise amplifier that has a large but finite spectral bandwidth. We referred to this model as the extended model. In Ref. [21] for the extended model the conditional PDF, the optimal input signal distribution, which maximizes the mutual information calculated in the leading order in the parameter 1/SNR was presented. These results were obtained both analytically and numerically. We vindicated the conclusions of the Ref. [14]; the informational characteristics in the extended model are significantly different from ones in the per-sample model.

In the present review we sum up the main results and the methods developed in our works [11,19,20,21] for the per-sample and extended nondispersive models. For the per-sample model we find the lower bound of the channel capacity and compare our results with ones obtained earlier. Attention is mainly given to the path integral approach since it can be applied for future considerations of the nonlinear channels with dispersion. The paper is organized as follows. In the next Section, we describe the per-sample and extended channel models. In Section 3 we consider the quantities which should be calculated to find the capacities of the per-sample and extended models. Then we consider in detail the conditional probability density function, entropies, mutual information, and the lower bound of the capacity for the per-sample model. After that, we proceed to consideration of the same quantities for the extended model. In conclusion, we discuss our results and possible applications of our methods to the nonlinear channels with non-zero dispersion.

## 2. Channel Models

### 2.1. Per-Sample Model

Following the papers [4,17] we imply that in the case of the per-sample model the equation of signal propagation has the form:(3)$$\begin{array}{cc} \; & \partial_{z}\psi \left(z\right)-i\gamma {\left|\psi \left(z\right)\right|}^{2}\psi \left(z\right)=\eta \left(z\right), \end{array}$$
where γ is the Kerr nonlinearity, ψ(z) is the signal function which obeys the boundary conditions ψ(0)=X, ψ(L)=Y, *L* is the signal propagation distance, *X* is the input signal and *Y* is the output signal. One can see that the Equation (Equation 3) does not contain terms which lead to decreasing of the signal in the propagation process and terms that compensate for the decreasing of the signal. It means that in the model, we have distributed amplifiers which completely compensate the attenuation of the signal in its propagation in the optical fiber. The only trace of amplifiers in Equation (Equation 3) is the nose function η(z). The functions η(z) has the following properties: the noise has zero mean ${\langle \eta \left(z\right)\rangle }_{\eta }=0$ and following correlation function ${\langle \eta \left(z\right)\bar{\eta }\left(z^{\prime }\right)\rangle }_{\eta }=Q\delta \left(z-z^{\prime }\right)\;$. Here and below the bar means the complex conjugation and δ(x) is the Dirac delta-function. The coefficient *Q* is the noise power per unit length, so QL is the noise power in the channel. The brackets ${\langle \ldots \rangle }_{\eta }$ mean the averaging over the noise realizations in the channel.

### 2.2. Extended Model

In the model (Equation 3) the bandwidths of the input signal, amplifiers and receiver cannot be taken into account since the functions ψ and η do not depend on time. In order to include these bandwidths we extend the previous model. In the extended model the signal ψ(z,t) does depend on time. The propagation is described by the stochastic NLSE with zero dispersion:(4)$$\begin{array}{cc} & \;\;\;\partial_{z}\psi \left(z,t\right)-i\gamma {\left|\psi \left(z,t\right)\right|}^{2}\psi \left(z,t\right)=\eta \left(z,t\right)\;, \end{array}$$
where the coefficient γ is also the Kerr nonlinearity coefficient. The function ψ(z,t) obeys the following conditions:(5)$$\begin{array}{c} \psi (0,t)=X(t)\;,\;\psi (L,t)=Y(t). \end{array}$$

The noise function η(z,t) has zero mean ${\langle \eta \left(z,t\right)\rangle }_{\eta }=0$. We also imply that the correlation function has finite bandwidth, so the correlator for the function
(6)$$\begin{array}{c} \eta \left(z,\omega \right)=\int_{-\infty }^{\infty }dte^{i\omega t}\eta \left(z,t\right) \end{array}$$
reads
(7)$$\begin{array}{ccc} {\langle \eta \left(z,\omega \right)\bar{\eta }\left(z^{\prime },\omega^{\prime }\right)\rangle }_{\eta } & = & 2\pi Q\delta \left(\omega -\omega^{\prime }\right)\theta \left(\frac{W^{\prime }}{2}-\left|\omega \right|\right)\delta \left(z-z^{\prime }\right)\;. \end{array}$$

Here, the parameter Q denotes a noise power per unit length and per unit frequency; θ(x) is the Heaviside theta-function. The theta-function $\theta \left(\frac{W^{\prime }}{2}-\left|\omega \right|\right)$ indicates that the noise is not zero within the interval $\left[-W^{\prime }/2,W^{\prime }/2\right]$, i.e., the bandwidth of the noise is equal to $W^{\prime }$. Performing the Fourier transform of Equation (Equation 7) we arrive at the correlator in the time domain:(8)$$\begin{array}{c} {\langle \eta \left(z,t\right)\bar{\eta }\left(z^{\prime },t^{\prime }\right)\rangle }_{\eta }=\frac{Q}{\pi (t-t^{\prime })}\sin\left(\frac{W^{\prime }\left(t-t^{\prime }\right)}{2}\right)\delta \left(z-z^{\prime }\right)\;. \end{array}$$

It is easy to check that if the time difference $t-t^{\prime }=2\pi n/W^{\prime }$, and *n* is integer, then the correlator (Equation 8) equals to zero. Therefore, the noise at times *t* and $t^{\prime }$ is not correlated, thus we can solve Equation (Equation 4) for different times $t_{j}=j\Delta$ independently. Here *j* is the integer number and $\Delta =2\pi /W^{\prime }$ is the grid spacing in the time domain. Therefore, instead of the continuous time model (Equation 4) we can consider the set of the discrete models:(9)$$\begin{array}{cc} & \;\;\;\partial_{z}\psi \left(z,t_{j}\right)-i\gamma {\left|\psi \left(z,t_{j}\right)\right|}^{2}\psi \left(z,t_{j}\right)=\eta \left(z,t_{j}\right)\; \end{array}$$
for the set of the time moments $t_{j}$. So we obtain the set of independent time channels, and instead of the continuous input and output conditions (Equation 5) we obtain the set of the discrete ones:(10)$$\begin{array}{c} \psi \left(0,t_{j}\right)=X\left(t_{j}\right)\;,\;\psi \left(L,t_{j}\right)=Y\left(t_{j}\right)\;. \end{array}$$

To include the bandwidth of the input signal to the model we represent the initial signal X(t) in the form:(11)$$\begin{array}{c} X\left(t\right)=\sum_{k=-N}^{N}C_{k}\;f\left(t-kT_{0}\right), \end{array}$$
where $C_{k}$ are complex random coefficients which carry the information. These coefficients have the probability density function $P_{X}\left[\left\{C\right\}\right]$, where $\left\{C\right\}=\left\{C_{-N},\ldots ,C_{N}\right\}$. We restrict our consideration by the envelopes f(t) which obey the following properties: the function f(t) is the real function and normalized by the condition: $\int_{-\infty }^{\infty }\frac{dt}{T_{0}}f^{2}\left(t\right)=1$; for integers *k* and *m* we have
(12)$$\begin{array}{c} \int_{-\infty }^{\infty }\frac{dt}{T_{0}}f\left(t-kT_{0}\right)f\left(t-mT_{0}\right)\approx 0,\;k\neq m. \end{array}$$

The last property means that the overlapping of the functions $f(t-kT_{0})$ and $f(t-mT_{0})$ is negligible. It means that we assume the smallness of the effects of the overlapping. The smallness of these effects will be discussed below. The function f(t) has almost finite support $\left[-T_{0}/2,T_{0}/2\right]$, and the input signal X(t) is defined on the interval $T=\left(2N+1\right)T_{0}$. The finiteness of the support $\left[-T_{0}/2,T_{0}/2\right]$ means that the frequency support of the function f(t) (and as a consequence, the support of the function X(t)) is infinite. However, we imply that
(13)$$\begin{array}{c} \int_{W}{\left|X\left(\omega \right)\right|}^{2}d\omega \approx \int_{W^{\prime }}{\left|X\left(\omega \right)\right|}^{2}d\omega \;, \end{array}$$
where X(ω) is the Fourier transformation of X(t). The relation (Equation 13) means that $T_{0}W\gg 1$. So we can say that the bandwidth of the input signal is *W*.

The bandwidth broadening of the signal propagating through optical fiber is associated with the nonlinearity and noise η. To estimate the broadening which is connected with nonlinearity we can find the solution $\Phi (z,t_{j})$ of Equation (Equation 9) with zero noise:(14)$$\begin{array}{cc} & \Phi \left(z,t\right)=X\left(t\right)e^{{i\gamma z|X\left(t\right)|}^{2}}. \end{array}$$

This solution obeys the input condition Φ(0,t)=X(t). Since we know the solution Φ(L,t), we can find its bandwidth $\tilde{W}$. Strictly speaking, the bandwidth $\tilde{W}$ is formally infinite, but the most part of the signal power is localized in the finite frequency region that can be specified as
(15)$$\begin{array}{cc} & \tilde{W}=\frac{\int_{-\infty }^{\infty }dt{\left|\partial_{t}\Phi \left(L,t\right)\right|}^{2}}{\int_{-\infty }^{\infty }dt{\left|\Phi (L,t)\right|}^{2}}. \end{array}$$

Below we assume the following hierarchy
(16)$$\begin{array}{c} W\leq \tilde{W}\ll W^{\prime }, \end{array}$$
where $\tilde{W}$ is the bandwidth of the function Φ(L,t).

To include the receiver bandwidth to the model we introduce the procedure of the output signal detection. In our model the receiver gets the information from the output signal, i.e., it recovers the coefficients {C}. We consider the following detection model. The receiver measures the output signal $\psi (L,t_{j})$ at the discrete time moments $t_{j}$ for j=−M,…,M−1. Here the quantity M=T/(2Δ) is the total number of the time samples. Since $T\gg T_{0}$, we have that M≫N. Such property of the receiver means that its time resolution coincides with the time discretization Δ. From Equation (Equation 16) it follows that $\Delta \ll 1/\tilde{W}$, therefore the receiver completely recovers the output signal in the noiseless case. Then the receiver removes the nonlinear phase
(17)$$\begin{array}{c} \tilde{X}\left(t_{j}\right)=\psi \left(L,t_{j}\right)e^{-i\gamma L|\psi \left(L,t_{j}\right){|}^{2}}, \end{array}$$
and obtains the recovered input signal $\tilde{X}\left(t\right)$. In fact, the procedure (Equation 17) means that we use the backward propagation procedure for the channel with zero dispersion. Finally, from the function $\tilde{X}\left(t\right)$ the receiver recovers the coefficients ${\tilde{C}}_{k}$ by projecting $\tilde{X}\left(t\right)$ on the basis functions $f(t-kT_{0})$:(18)$$\begin{array}{c} {\tilde{C}}_{k}=\frac{1}{T_{0}}\int_{-\infty }^{\infty }dtf\left(t-kT_{0}\right)\tilde{X}\left(t\right)\approx \frac{\Delta }{T_{0}}\sum_{j=-M}^{M-1}f\left(t_{j}-kT_{0}\right)\tilde{X}\left(t_{j}\right). \end{array}$$

So the extended model contains the bandwidth of the input signal *W*, bandwidth the noise of amplifiers $W^{\prime }$, and bandwidth of the receiver. In our case, the bandwidth of the receiver coincides with the bandwidth of the noise because we choose the discretization in the information extracting procedure (Equation 18) coinciding with the initial channel discretization.

## 3. Channel Capacity and Its Bound

It is known [1] that the statistical properties of the memoryless channels such as the conditional entropy H[Y|X] and the output signal entropy H[Y] can be expressed through the conditional probability density function as
(19)$$\begin{array}{ccc} \;\;\;H[Y|X] & = & -\int DXDYP_{X}\left[X\right]P[Y|X]\log P\left[Y|X\right], \end{array}$$
(20)$$\begin{array}{ccc} \;\;\;H[Y] & = & -\int DYP_{out}\left[Y\right]\log P_{out}\left[Y\right], \end{array}$$
where P[Y|X] is the conditional probability density function (i.e., the probability density to receive the output signal *Y* for transmitted signal *X*), $P_{out}\left[Y\right]$ is the probability density function of the output signal. The distribution $P_{out}\left[Y\right]$ has the following form:(21)$$\begin{array}{ccc} \;\;\;P_{out}\left[Y\right] & = & \int DXP_{X}\left[X\right]P\left[Y|X\right]. \end{array}$$

The measures DX and DY are defined in such a way that
(22)$$\begin{array}{c} \int DXP_{X}\left[X\right]=1, \end{array}$$
(23)$$\begin{array}{c} \int DYP[Y|X]=1. \end{array}$$

The mutual information of a memoryless channel is defined through the entropy H[Y] of the output signal and the conditional entropy H[Y|X] as
(24)$$\begin{array}{c} I_{P_{X}\left[X\right]}=H\left[Y\right]-H\left[Y|X\right]. \end{array}$$

The channel capacity *C* is defined as the maximum of the functional $I_{P_{X}\left[X\right]}$ with respect to the input signal distribution $P_{X}\left[X\right]$:(25)$$\begin{array}{c} C=\max_{P_{X}\left[X\right]}I_{P_{X}\left[X\right]}. \end{array}$$

The maximum value of $I_{P_{X}\left[X\right]}$ should be calculated for the fixed average signal power. Note that since in Equations (Equation 19) and (Equation 20) we use the logarithm to the base *e*, here and below we measure the mutual information and the capacity in units nat/symbol. For the case of the per-sample channel, the signal power reads
(26)$$\begin{array}{c} P=\int DXP_{X}\left[X\right]{\left|X\right|}^{2}. \end{array}$$

For the case of the extended model the signal depends on time, therefore, we have
(27)$$\begin{array}{c} P=\int DX\left(t\right)P_{X}\left[X\right]\int_{-\infty }^{\infty }\frac{dt}{T}{\left|X\left(t\right)\right|}^{2}. \end{array}$$

So, to find the channel capacity we should know the conditional PDF for both models. Let us start with the calculation of the conditional PDF for the per-sample model.

### 3.1. Per-Sample Model

#### 3.1.1. Conditional Probability Density Function

The conditional PDF can be written in the form of the path integral over all realizations of the signal ψ(z) in the channel [17,18]:(28)$$\begin{array}{cc} & P\left[Y|X\right]=\int_{\psi (0)=X}^{\psi (L)=Y}\;\;\;\;D\psi \;e^{-S[\psi ]/Q}\;, \end{array}$$
where the effective action S[ψ] reads as the integral of the squared left-hand side of Equation (Equation 3) over the variable *z*: $S\left[\psi \right]=\int_{0}^{L}dz|\partial_{z}\psi -i\gamma {\left|\psi \right|}^{2}\psi {|}^{2}$. To calculate the path integral, we use the retarded discretization scheme which reflects the physics of the propagation process. The retarded discretization assumes that the derivation means $\left(\partial_{z}\psi \right)\left(z_{n}\right)=\left(\psi \left(z_{n}\right)-\psi \left(z_{n-1}\right)\right)/\Delta_{z}$, where $z_{n}=n\Delta_{z}$, $\Delta_{z}=L/N_{z}$ ($z_{0}=0$, $z_{N_{z}}=L$), and any integral $\int_{0}^{L}dzf\left(z\right)$ means $\Delta_{z}\sum_{n=0}^{N_{z}}f\left(z_{n}\right)$. The general approach for the derivation of the representation (Equation 28) and the argumentation for the retarded scheme one can find in Ref. [24].

For the first time, the conditional PDF P[Y|X] for the per-sample model was obtained in the form of infinite series in Ref. [4]:(29)$$\begin{array}{c} P\left[Y|X\right]=\frac{1}{\pi Q}\sum_{m=-\infty }^{+\infty }\;e^{im(\phi^{\left(Y\right)}-\phi^{\left(X\right)}-\mu )}\frac{\exp\{-\frac{\rho^{2}+\rho^{\prime 2}}{Q}k_{m}\coth\left(k_{m}L\right)\}}{\sinh(k_{m}L)}k_{m}I_{\left|m\right|}\left(\frac{2k_{m}\rho \rho^{\prime }}{Q\sinh(k_{m}L)}\right), \end{array}$$
where the input signal $X=\rho e^{i\phi^{\left(X\right)}}$, the output signal $Y=\rho^{\prime }e^{i\phi^{\left(Y\right)}}$, $\mu ={\gamma L|X|}^{2}$ is a dimensionless nonlinear parameter, $k_{m}=\sqrt{Q\;m\gamma }e^{i\frac{\pi }{4}}$, and $I_{\left|m\right|}\left(z\right)$ is the modified Bessel function of the index |m|. The representation (Equation 29) was rederived using the path integral representation (Equation 28) in Refs. [17,18]. In Ref. [18] the authors applied two various methods: the recursive derivation based on the discretizing of the nondispersive NLSE and the properties of Markov chains (Chapman–Kolmogorov equation); and the derivation of the conditional PDF via the stochastic approach and Ito calculus [25].

Using the result (Equation 29), the lower bound of the capacity for the large signal power was found in Ref. [17]:(30)$$\begin{array}{c} C\geq \frac{\log\left(\mathrm{SNR}\right)}{2}+O\left(1\right). \end{array}$$

Here the signal-to-noise ratio has the form
(31)$$\begin{array}{c} \mathrm{SNR}=\frac{P}{QL}. \end{array}$$

Recall that the noise power for the per-sample model is QL. To obtain the result (Equation 30) the authors demonstrate that at large-signal power the phase of the output signal occupies the entire phase interval [0,2π] due to the interaction of the signal with the noise. As a result, the phase does not carry the information, see also [18]. Therefore, to find the lower bound for the large signal power it is necessary to take only the term with m=0 in Equation (Equation 29).

At the so-called intermediate power *P* range:(32)$$\begin{array}{c} QL\ll P\ll {\left(Q\gamma^{2}L^{3}\right)}^{-1}, \end{array}$$
it is necessary to take into account terms with m≠0 in the Equation (Equation 29). In Ref. [18], using Equation (Equation 29), the attempt was made to find the lower bound for the capacity in the intermediate power range. However due to the inconvenience of using of Equation (Equation 29) for analytical calculation this attempt was unsuccessful.

To calculate the mutual information in the intermediate power range we have to find the conditional probability density function (Equation 29) in a more convenient form. In Ref. [19] we found the method of the conditional probability density function calculation in the form of expansion in the parameter 1/SNR. Using the developed method we found the conditional PDF in the convenient for further calculation form. The expansion in 1/SNR, or in the small parameter *Q* similar to the semi-classical approximation (expansion in small Planck’s constant *ℏ*) in quantum mechanics.

Using the “semi-classical” method, see Ref. [26], we perform the change of integration variables (the simple shift with the Jacobian equals to unity) in Equation (Equation 28):(33)$$\begin{array}{c} \psi \left(z\right)=\Psi_{cl}\left(z\right)+\tilde{\psi }\left(z\right), \end{array}$$
and represent the conditional PDF (Equation 28) in the form:(34)$$\begin{array}{c} P\left[Y|X\right]=\Lambda \;e^{-S[\Psi_{cl}\left(z\right)]/Q}\;, \end{array}$$
where the normalization factor Λ has the form
(35)$$\begin{array}{c} \Lambda =\int_{\tilde{\psi }\left(0\right)=0}^{\tilde{\psi }\left(L\right)=0}\;\;\;\;\;D\tilde{\psi }\;\exp\left\{-\frac{S\left[\Psi_{cl}\left(z\right)+\tilde{\psi }\left(z\right)\right]-S\left[\Psi_{cl}\left(z\right)\right]}{Q}\right\}, \end{array}$$
the function $\Psi_{cl}\left(z\right)$ is the “classical” solution of the equation $\delta S[\Psi_{cl}]=0$ (here δS is the variation of the action S[ψ]). The measure $D\tilde{\psi }$ for new variables $\tilde{\psi }$ in Equation (Equation 34) is defined as
(36)$$\begin{array}{c} D\tilde{\psi }={\left(\frac{1}{\Delta_{z}\pi Q}\right)}^{N_{z}}\prod_{i=1}^{N_{z}-1}dRe{\tilde{\psi }}_{i}\;dIm{\tilde{\psi }}_{i}, \end{array}$$
here ${\tilde{\psi }}_{i}=\tilde{\psi }\left(z_{i}\right)$, $\Delta_{z}=\frac{L}{N_{z}}$ is the grid spacing.

The Euler–Lagrange equation $\delta S[\Psi_{cl}]=0$ has the following explicit form
(37)$$\begin{array}{c} \frac{d^{2}\Psi_{cl}}{dz^{2}}-4i\gamma {\left|\Psi_{cl}\right|}^{2}\frac{d\Psi_{cl}}{dz}-3\gamma^{2}{\left|\Psi_{cl}\right|}^{4}\Psi_{cl}=0. \end{array}$$

The boundary conditions for the function $\Psi_{cl}\left(z\right)$ are as follows
(38)$$\begin{array}{c} \Psi_{cl}\left(0\right)=X,\;\Psi_{cl}\left(L\right)=Y. \end{array}$$

Equation (Equation 37) is of great concern and we present two different approaches to its solution.

One can find the analytical solution of Equation (Equation 37) in the polar coordinate system [19]: $\Psi_{cl}\left(z\right)=\rho \left(\zeta \right)e^{i\theta (\zeta )}$, ζ=z/L. The solution depends on the following real integration constants: *E*, $\tilde{\mu }$, $\zeta_{0}$ and $\theta_{0}$. These constants should be expressed from two boundary conditions (Equation 38). There are two different types of the solution. The first and the second type corresponds to the cases E≥0 and E≤0, respectively. In the first case we have the solution:(39)$$\begin{array}{c} \rho^{2}\left(\zeta \right)=\frac{1}{2L\gamma }\left(\tilde{\mu }+\sqrt{{\tilde{\mu }}^{2}-k^{2}}\cos\left[2k\left(\zeta -\zeta_{0}\right)\right]\right)\;, \end{array}$$
(40)$$\begin{array}{cc} \theta (\zeta )= & \frac{\tilde{\mu }}{2}\left(\zeta -\zeta_{0}\right)+\sqrt{{\tilde{\mu }}^{2}-k^{2}}\;\frac{\sin[2k\left(\zeta -\zeta_{0}\right)]}{4k}+\\ & \arctan\left[\left(\tilde{\mu }-\sqrt{{\tilde{\mu }}^{2}-k^{2}}\right)\frac{\tan[k\left(\zeta -\zeta_{0}\right)]}{k}\right]+\theta_{0}, \end{array}$$
where $k=\sqrt{2E}$. The solution (Equation 39), (Equation 40) is obtained under conditions $\tilde{\mu }\geq k\geq 0$. The integration constants $\tilde{\mu }$, *k* and $\zeta_{0}$ must be found from the boundary conditions: (41)$$\begin{array}{c} {\left|X\right|}^{2}=\rho^{2}\left(0\right)=\frac{\tilde{\mu }+\sqrt{{\tilde{\mu }}^{2}-k^{2}}\cos\left[2k\zeta_{0}\right]}{2L\gamma }, \end{array}$$(42)$$\begin{array}{c} {\left|Y\right|}^{2}=\rho^{2}\left(1\right)=\frac{\tilde{\mu }+\sqrt{{\tilde{\mu }}^{2}-k^{2}}\cos\left[2k\left(1-\zeta_{0}\right)\right]}{2L\gamma }, \end{array}$$(43)$$\begin{array}{c} \phi^{\left(X\right)}=\theta \left(0\right)=-\frac{\tilde{\mu }}{2}\zeta_{0}-\sqrt{{\tilde{\mu }}^{2}-k^{2}}\;\frac{\sin[2k\zeta_{0}]}{4k}-\\ \arctan\left[\left(\tilde{\mu }-\sqrt{{\tilde{\mu }}^{2}-k^{2}}\right)\frac{\tan[k\zeta_{0}]}{k}\right]+\theta_{0}, \end{array}$$(44)$$\begin{array}{c} \phi^{\left(Y\right)}=\theta \left(1\right)=\frac{\tilde{\mu }}{2}\left(1-\zeta_{0}\right)+\sqrt{{\tilde{\mu }}^{2}-k^{2}}\;\frac{\sin[2k\left(1-\zeta_{0}\right)]}{4k}+\\ \arctan\left[\left(\tilde{\mu }-\sqrt{{\tilde{\mu }}^{2}-k^{2}}\right)\frac{\tan[k\left(1-\zeta_{0}\right)]}{k}\right]+\theta_{0}. \end{array}$$

If the integration constants are found then we can express the action in the form:(45)$$\begin{array}{c} S\left[\Psi_{cl}\left(z;E,\tilde{\mu },\zeta_{0},\theta_{0}\right)\right]=\frac{k^{2}}{2\gamma L}\left(\tilde{\mu }-\sqrt{{\tilde{\mu }}^{2}-k^{2}}\;\frac{\sin\left[2k\left(1-\zeta_{0}\right)\right]+\sin\left[2k\zeta_{0}\right]}{2k}\right). \end{array}$$

In the second case (E≤0) the solution has the form
(46)$$\begin{array}{c} \rho^{2}\left(\zeta \right)=\frac{-\tilde{\mu }+\sqrt{{\tilde{\mu }}^{2}+k^{2}}\cosh\left[2k\left(\zeta -\zeta_{0}\right)\right]}{2L\gamma }\;, \end{array}$$
(47)$$\begin{array}{cc} \theta (\zeta )= & -\frac{\tilde{\mu }}{2}\left(\zeta -\zeta_{0}\right)+\sqrt{{\tilde{\mu }}^{2}+k^{2}}\;\frac{\sinh[2k\left(\zeta -\zeta_{0}\right)]}{4k}-\\ & \arctan\left[\left(\tilde{\mu }+\sqrt{{\tilde{\mu }}^{2}+k^{2}}\right)\frac{\tanh[k\left(\zeta -\zeta_{0}\right)]}{k}\right]+\theta_{0}, \end{array}$$
where $k=\sqrt{-2E}$. The integration parameters $\tilde{\mu }$, *k*, $\zeta_{0}$, and $\theta_{0}$ are derived from the same procedure as in the first case. The action has the form
(48)$$\begin{array}{c} S\left[\Psi_{cl}\left(z;E,\tilde{\mu },\zeta_{0},\theta_{0}\right)\right]=\frac{k^{2}}{2\gamma L}\left(\tilde{\mu }+\sqrt{{\tilde{\mu }}^{2}+k^{2}}\;\frac{\sinh\left[2k\left(1-\zeta_{0}\right)\right]+\sinh\left[2k\zeta_{0}\right]}{2k}\right). \end{array}$$

One can see that to obtain the solution ρ(ζ), θ(ζ), it is necessary to solve the system of nonlinear equations Equations (Equation 41)–(Equation 44) for integration constants. Of course, the system (Equation 41)–(Equation 44) can be solved numerically, but for the analytical calculation of the mutual information, it is necessary to develop the method which allows us to find the solution $\Psi_{cl}\left(z\right)$ and action $S[\Psi_{cl}\left(z\right)]$ analytically as a functional of the input *X* and output *Y* signals.

In Ref. [19] we have proposed the method based on the expansion of the semi-classical solution in the vicinity of the solution of the Equation (Equation 3) with zero noise. This makes sense when the noise power is much less than the signal power.

To demonstrate the approach, we find the solution of (Equation 37) in the leading order in the parameter 1/SNR, linearizing Equation (Equation 37) in the vicinity of the solution $\Psi_{0}\left(z\right)$. The function $\Psi_{0}\left(z\right)$ is the solution of the Equation (Equation 3) with zero noise. It obeys the input boundary condition $\Psi_{0}\left(0\right)=X=\rho e^{i\phi^{\left(X\right)}}$, ρ=|X|. Note that we do not assume smallness of the nonlinearity. The function $\Psi_{0}\left(z\right)$ reads
(49)$$\begin{array}{c} \Psi_{0}\left(z\right)=\rho \exp\left\{i\mu \frac{z}{L}+i\phi^{\left(X\right)}\right\}, \end{array}$$
where $\mu =\gamma L\rho^{2}$ is the nonlinear dimensionless parameter. Let us represent the “classical” solution $\Psi_{cl}\left(z\right)$ in the form
(50)$$\begin{array}{c} \Psi_{cl}\left(z\right)=\Psi_{0}\left(z\right)+\delta \Psi \left(z\right). \end{array}$$

Here
(51)$$\begin{array}{c} \delta \Psi \left(z\right)=\varkappa \left(z\right)\exp\left\{i\mu \frac{z}{L}+i\phi^{\left(X\right)}\right\}, \end{array}$$
where the function ϰ(z) is assumed to be much less than the ρ: |ϰ(z)|≪ρ, i.e., $\Psi_{cl}\left(z\right)$ is close to $\Psi_{0}\left(z\right)$. Note that the function ϰ(z) depends on the output boundary conditions, therefore in a general case, the ratio |ϰ(z)|/ρ can be of order of unity. The boundary conditions for the function ϰ(z) are as follows:(52)$$\begin{array}{c} \varkappa \left(0\right)=0,\;\varkappa \left(L\right)=Ye^{-i\phi^{\left(X\right)}-i\mu }-\rho \equiv x_{0}+iy_{0}, \end{array}$$
where $x_{0}=Re\left\{\varkappa \left(L\right)\right\}$ and $y_{0}=Im\left\{\varkappa \left(L\right)\right\}$. It is important, that the configurations of ϰ(z) at which $\Psi_{cl}\left(z\right)$ significantly deviates from $\Psi_{0}\left(z\right)$ are statistically irrelevant.

One can check that since the action achieves the absolute minimum ($S[\Psi_{0}\left(z\right)]=0$) on the solution $\Psi_{0}\left(z\right)$, the expansion of the action $S[\Psi_{0}\left(z\right)+\delta \Psi \left(z\right)]$ starts from the quadratic terms for small ϰ(z):$$S\left[\Psi_{0}\left(z\right)+\delta \Psi \left(z\right)\right]\propto \varkappa^{2}\left(z\right).$$

Therefore, the exponent $e^{-S[\Psi_{cl}\left(z\right)]/Q}$ and, as a result, the conditional PDF P[Y|X] decreases exponentially when $\varkappa \left(z\right)\gg \sqrt{QL}$.

The next step in evaluation of the conditional probability P[Y|X] is the calculation of the path integral (Equation 35). To calculate the integral (Equation 35) in the leading order in parameter $1/\sqrt{\mathrm{SNR}}$ one should retain only terms quadratic in the function $\tilde{\psi }$ in the integrand. Any extra powers of $\tilde{\psi }$ or ϰ lead to suppression in the multiplicative parameter $\sqrt{QL}$, since for small *Q* the dominant contribution to the path integral comes from region where $\tilde{\psi }∼\sqrt{QL}$. Since we calculate the the integral (Equation 35) in the leading order in the parameter *Q*, the function $\Psi_{cl}\left(z\right)$ can be replaced by the function $\Psi_{0}\left(z\right)$ in the exponent $\exp\left\{-(S\left[\Psi_{cl}\left(z\right)+\tilde{\psi }\left(z\right)\right]-S\left[\Psi_{cl}\left(z\right)\right])/Q\right\}$.

To find the next-to-leading order corrections in the parameter $1/\sqrt{\mathrm{SNR}}$ to the conditional PDF P[Y|X] one should keep both ϰ(z) in $\Psi_{cl}\left(z\right)$ and higher powers of $\tilde{\psi }$ in the exponent in Equation (Equation 35). Details of the path integral calculation in the leading and next-to-leading orders in $1/\sqrt{\mathrm{SNR}}$ one can find in Ref. [19]. Here we present only the result obtained in the leading order in the parameter $1/\sqrt{\mathrm{SNR}}$:(53)$$\begin{array}{cc} & \;\;\;P\left[Y|X\right]=\frac{\exp\left\{-\frac{\left(1+4\mu^{2}/3\right)x_{0}^{2}-2\mu x_{0}y_{0}+y_{0}^{2}}{QL(1+\mu^{2}/3)}\right\}}{\pi QL\sqrt{1+\mu^{2}/3}}, \end{array}$$
where $x_{0}=Re\left\{Ye^{-i\phi^{\left(X\right)}-i\mu }-\rho \right\}$ and $y_{0}=Im\left\{Ye^{-i\phi^{\left(X\right)}-i\mu }-\rho \right\}$. One can see that the expression (Equation 53) is much simpler than the exact result (Equation 29). Note that the distribution (Equation 53) has the following property
(54)$$\begin{array}{cc} & \lim_{Q\to 0}P\left[Y|X\right]=\delta \left(Y-\Psi_{0}\left(L\right)\right). \end{array}$$

The limit (Equation 54) is the deterministic limit of P[Y|X] in the absence of noise. Also Equation (Equation 53) has the correct limit for small γ:(55)$$\begin{array}{c} \lim_{\gamma \to 0}P\left[Y|X\right]=\frac{e^{{-|Y-X|}^{2}/QL}}{\pi QL}\;, \end{array}$$
where the right-hand-side is the conditional PDF for the linear nondispersive channel with the additive noise.

Let us compare the result obtained in the leading in $1/\sqrt{\mathrm{SNR}}$ order (Equation 53) with exact result (Equation 29). In Figure 1 we plot the PDF P[Y|X] as the function of |Y| for $X=2\;{\mathrm{mW}}^{1/2}$, arg(Y)=μ, $\mu ={\gamma L|X|}^{2}=4$, so we choose $\gamma L={\mathrm{mW}}^{-1}$, and for two values of parameter QL: QL=1/2mW, QL=1/25mW (it corresponds to SNR=8 and SNR=100, respectively).

One can see the good agreement between the exact result (Equation 29) and the approximation (Equation 53) even for SNR = 8. For SNR=100 the approximation almost coincides with the exact result. In the case when the $\mathrm{SNR}\approx 10^{2}$–$10^{4}$, which corresponds to optical fiber channels, the difference between the approximation and the exact result is of the order of $1/\sqrt{\mathrm{SNR}}$. To decrease this difference we should calculate the corrections of the order of $1/\sqrt{\mathrm{SNR}}$ and 1/SNR, see the Section 3.1.3.

#### 3.1.2. Probability Density Function $P_{out}\left[Y\right]$

Let us consider the integral (Equation 21) which defines the the probability density function of the output signal:(56)$$\begin{array}{c} P_{out}\left[Y\right]=\int DXP\left[Y|X\right]P_{X}\left[X\right], \end{array}$$
where the distribution $P_{X}\left[X\right]$ is a smooth function. Since the input signal power is *P*, we can expect that the function $P_{X}\left[X\right]$ changes on the scale $X∼\sqrt{P}$ which is assumed to be much greater than $\sqrt{QL}$. For this case we can use the Laplace’s method [27] to calculate the integral (Equation 56) up to the terms proportional to the noise power QL, for details of the calculation see Appendix C in Ref. [19]. The idea of the integral (Equation 56) calculation is based on the fact that the function P[Y|X] is much narrower than the function $P_{X}\left[X\right]$ (the function P[Y|X] is almost Dirac delta-function for the function $P_{X}\left[X\right]$). Therefore, the calculation of the integral is simple and the result has the form [19]:(57)$$\begin{array}{c} P_{out}\left[Y\right]=\int DXP\left[Y|X\right]P_{X}\left[X\right]\approx P_{X}\left[Ye^{-{i\gamma |Y|}^{2}L}\right]. \end{array}$$

When obtaining the result (Equation 57) it is not required to pass to the limit Q→0 but only the relation P≫QL between the scales *P* and QL is used.

For the class of the distributions $P_{X}\left[X\right]$ depending only on absolute value |X| we have $P_{out}\left[Y\right]=P_{X}\left[|Y|\right]$. For such distributions we can calculate corrections to (Equation 57) in the parameter QL in any order in QL.

Let us restrict our consideration in the remainder of this sub-subsection to the case of the distributions $P_{X}\left[X\right]$ depending only on |X|. We can use conditional PDF P[Y|X] (Equation 29) found in Ref. [17]. In this case the function $P_{out}\left[Y\right]$ depends only on $\left|Y\right|=\rho^{\prime }$:(58)$$\begin{array}{c} P_{out}\left[\rho^{\prime }\right]=\frac{2e^{-\rho^{\prime 2}/\left(QL\right)}}{QL}\int_{0}^{\infty }d\rho \rho e^{-\rho^{2}/\left(QL\right)}I_{0}\left(\frac{2\rho \rho^{\prime }}{QL}\right)P_{X}\left[\rho \right]. \end{array}$$

Using this formula one can obtain the simple relation for $P_{out}\left[\rho^{\prime }\right]$ within the perturbation theory in the parameter QL. Performing the zero order Hankel transformation [27]:(59)$$\begin{array}{c} \hat{P}\left[k\right]=\int_{0}^{\infty }d\rho \rho J_{0}\left(k\rho \right)P_{X}\left[\rho \right] \end{array}$$
for both sides of Equation (Equation 58), and using the standard integral with the Bessel $J_{\nu }\left(x\right)$ and modified Bessel functions [28]
$$\begin{array}{c} \int_{0}^{\infty }dzze^{-pz^{2}}J_{\nu }\left(bz\right)I_{\nu }\left(cz\right)=\frac{1}{2p}J_{\nu }\left(\frac{bc}{2p}\right)e^{\frac{c^{2}-b^{2}}{4p}}, \end{array}$$
we arrive at the relation between the Hankel images of the output and input signal PDFs:(60)$$\begin{array}{c} {\hat{P}}_{out}\left[k\right]=e^{-k^{2}\frac{QL}{4}}\hat{P}\left[k\right]. \end{array}$$

Then, we perform the inverse Hankel transformation
(61)$$\begin{array}{c} P_{X}\left[\rho \right]=\int_{0}^{\infty }dkkJ_{0}\left(k\rho \right)\hat{P}\left[k\right], \end{array}$$
and obtain the following important result
(62)$$\begin{array}{c} P_{out}\left[\rho \right]=e^{\frac{QL}{4}\Delta_{\rho }}P_{X}\left[\rho \right], \end{array}$$
where $\Delta_{\rho }=\frac{d^{2}}{d\rho^{2}}+\frac{1}{\rho }\frac{d}{d\rho }$ is the two-dimensional radial Laplace operator. The relation (Equation 62) allows us to find the corrections of orders of ${\left(QL\right)}^{n}$ to $P_{out}\left[\rho \right]$ by the expansion of the exponent and calculation of the action of the differential operator $\Delta_{\rho }^{n}$ on the input PDF $P_{X}\left[\rho \right]$.

Let us consider the widely used example of the modified Gaussian distribution with one parameter β:(63)$$\begin{array}{c} P_{X}^{\left(\beta \right)}\left[X\right]=\frac{\exp\left\{-{\beta |X|}^{2}/\left(2P\right)\right\}{\left|X\right|}^{\beta -2}}{\pi \Gamma \left(\beta /2\right){\left(2P/\beta \right)}^{\beta /2}}, \end{array}$$
where $\Gamma \left(x\right)$ is the Euler gamma function. In the case of β>0 the distribution $P_{X}^{\left(\beta \right)}\left[X\right]$ obeys the conditions: $2\pi \int_{0}^{\infty }d\rho \rho P_{X}^{\left(\beta \right)}\left[\rho \right]=1$, $2\pi \int_{0}^{\infty }d\rho \rho^{3}P_{X}^{\left(\beta \right)}\left[\rho \right]=P$. The last one means that the average power of the input signal is equal to *P*. The generalized distribution $P_{X}^{\left(\beta \right)}\left[X\right]$ for β=1 is called as the half-Gaussian distribution
(64)$$\begin{array}{c} P_{X}^{\left(1\right)}\left[X\right]=\frac{\exp\left\{-{\left|X\right|}^{2}/\left(2P\right)\right\}}{\pi \sqrt{2\pi P}\left|X\right|},\; \end{array}$$
and the Gaussian distribution for β=2:(65)$$\begin{array}{cc} & P_{X}^{\left(2\right)}\left[X\right]=\frac{e^{-{\left|X\right|}^{2}/P}}{\pi P}. \end{array}$$

Inserting (Equation 63) into Equation (Equation 58) we arrive at the standard integral, see [28], and obtain:(66)$$\begin{array}{c} P_{out}^{\left(\beta \right)}\left[Y\right]={}_{1}F_{1}\left(\frac{\beta }{2};1;\frac{{\left|Y\right|}^{2}2P}{QL(2P+\beta QL)}\right)\frac{\exp\{-|Y{|}^{2}/QL\}}{\pi QL}{\left(\frac{\beta QL}{2P+\beta QL}\right)}^{\beta /2}, \end{array}$$
where ${}_{1}F_{1}\left(\beta /2;1;z\right)$ is the confluent hypergeometric function. The function reduces to $e^{z}$ for the case of the Gaussian distribution, and to the expression $e^{z/2}I_{0}\left(z/2\right)$ for the case of the half-Gaussian one:(67)$$\begin{array}{cc} & P_{out}^{\left(2\right)}\left[Y\right]=\frac{e^{-{\left|Y\right|}^{2}/\left(P+QL\right)}}{\pi (P+QL)}, \end{array}$$
(68)$$\begin{array}{c} P_{out}^{\left(1\right)}\left[Y\right]=\frac{1}{\pi \sqrt{QL(2P+QL)}}I_{0}\left(\;\;\frac{{\left|Y\right|}^{2}P}{QL(2P+QL)}\right)\exp\left\{-\frac{{\left|Y\right|}^{2}\left(P+QL\right)}{QL(2P+QL)}\right\}. \end{array}$$

The result (Equation 57) can be obtained from Equation (Equation 68) in the case $QL\ll {\left|Y\right|}^{2}∼P$. Expanding the right-hand side of the Equation (Equation 67) in the parameter QL/P we obtain
(69)$$\begin{array}{cc} & P_{out}^{\left(1\right)}\left[Y\right]\approx P_{X}^{\left(1\right)}\left[|Y|\right] \end{array}$$
with accuracy O(QL). The result (Equation 69) coincides with the general relation (Equation 57).

#### 3.1.3. Lower Bound for the Channel Capacity

The estimates for the capacity of the per-sample model in the regime of very large SNR were obtained in Ref. [17]. To specify, it was considered the case $P\gg {\left(P_{noise}\gamma^{2}L^{2}\right)}^{-1}$, where $P_{noise}=QL$ is the noise power. In Ref. [17] the lower bound for the capacity of the per-sample channel was found. Using the trial half-Gaussian input signal PDF (Equation 64) authors obtained the following result:(70)$$\begin{array}{c} C\geq \frac{\log\left(\mathrm{SNR}\right)}{2}+\;\frac{1+\gamma_{E}-\log\left(4\pi \right)}{2}+O\;\left(\frac{\log(\mathrm{SNR})}{\mathrm{SNR}}\right), \end{array}$$
where $\gamma_{E}\approx 0.5772$ is the Euler constant. The second term on the right-hand side of Equation (Equation 30) was presented as O(1) in Ref. [17] but it can be found using Equations (23) and (24) of Ref. [17]. Comparing the result (Equation 70) with the Shannon result (Equation 1) it is worth noting that the pre-logarithmic factor 1/2 differs from the unity in the Shannon result. The physical meaning of the difference is that the signal’s phase does not carry information when the power is very large: $P\gg {\left(P_{noise}\gamma^{2}L^{2}\right)}^{-1}$, see Ref. [17].

The most interesting power regime for the per-sample model is so-called intermediate power range defined in (Equation 32). In the regime we have on the one hand the parameter SNR is large, $P_{noise}\ll P$, and on the other hand the signal-dependent phase does not yet occupy the entire phase interval [0,2π] due to the signal-to-noise interaction, i.e., the phase still carries information.

Capacity estimates in the intermediate power range (Equation 32) were presented in Ref. [18]. For such a power *P* the authors of this paper used the half-Gaussian input signal PDF (Equation 64) for the estimate of the lower bound for the capacity as well, see inequality (40) in Ref. [18]. But there were some flaws in the derivation of this inequality in Ref. [18], see the discussion in the Introduction of Ref. [19]. In our approach presented in Ref. [19] we solved a variational problem for the mutual information (Equation 24) with the normalization (Equation 22) and power restriction (Equation 26): we found both the optimal input signal distribution $P_{X}\left[X\right]$ maximizing the mutual information (Equation 24) in the leading and next-to-leading orders in $1/\sqrt{\mathrm{SNR}}$ in the intermediate power range. Let us proceed with this calculation.

To begin with, when the parameter SNR≫1 we can calculate the output signal entropy H[Y] by substituting $P_{X}\left[Y\exp\left\{-{i\gamma |Y|}^{2}L\right\}\right]$ to Equation (Equation 20) instead of $P_{out}\left[Y\right]$ due to the relation (Equation 57), and then performing the change of the integration variable $\phi =\phi^{\left(Y\right)}+\gamma {\left|Y\right|}^{2}L$ we obtain
(71)$$\begin{array}{c} \;\;H\left[Y\right]\approx -\int_{0}^{2\pi }d\phi \;\;\int_{0}^{\infty }d\rho^{\prime }\rho^{\prime }P_{X}\left[\rho^{\prime }e^{i\phi }\right]\log P_{X}\left[\rho^{\prime }e^{i\phi }\right]. \end{array}$$

Note that the output signal entropy in the form (Equation 71) coincides with the input signal entropy H[X] with the accuracy O(QL).

Secondly, to obtain the conditional entropy H[Y|X] we substitute the conditional PDF P[Y|X] in the form of Equation (Equation 53) into Equation (Equation 19). Then we change the integration variables DY≡dReYdImY to $dx_{0}dy_{0}$, and perform the integration over $x_{0}$, $y_{0}$. The result has the form:(72)$$\begin{array}{c} H\left[Y|X\right]=\log\left(e\pi QL\right)+\int_{0}^{2\pi }d\phi \int_{0}^{\infty }d\rho \;\rho \;P_{X}\left[\;\rho \;e^{i\phi }\right]\log\sqrt{1+\gamma^{2}L^{2}\rho^{4}/3}. \end{array}$$

Thirdly, to find the optimal input signal distribution $P_{X}^{opt}\left[X\right]$ we solve the variational problem for the functional $J[P_{X},\lambda_{1},\lambda_{2}]$:(73)$$\begin{array}{c} J\left[P_{X},\lambda_{1},\lambda_{2}\right]=H\left[Y\right]-H\left[Y|X\right]-\lambda_{1}\left(\int DXP_{X}\left[X\right]-1\right)-\\ \lambda_{2}\left(\int DXP_{X}\left[X\right]{\left|X\right|}^{2}-P\right), \end{array}$$
where $\lambda_{1,2}$ are Lagrange multipliers which corresponds to the normalization condition (Equation 22) and the condition (Equation 26) of the fixed average signal power *P*. The solution $P_{X}^{opt}\left[X\right]$ of the corresponding Euler–Lagrange equations for (Equation 73) referred to as the “optimal” distribution:(74)$$\begin{array}{cc} & P_{X}^{opt}\left[X\right]=N_{0}\left(P\right)\frac{e^{-\lambda_{0}\left(P\right){\left|X\right|}^{2}}}{\sqrt{1+\gamma^{2}L^{2}{\left|X\right|}^{4}/3}}, \end{array}$$
where coefficients $N_{0}\left(P\right)$ and $\lambda_{0}\left(P\right)$ are determined from the conditions: (75)$$\begin{array}{cc} & \int DXP_{X}^{opt}\left[X\right]=2\pi N_{0}\left(P\right)\int_{0}^{\infty }\frac{d\rho \;\rho \;e^{-\lambda_{0}\left(P\right)\rho^{2}}}{\sqrt{1+\gamma^{2}L^{2}\rho^{4}/3}}=1, \end{array}$$(76)$$\begin{array}{c} \;\;\;\;\int \;\;DXP_{X}^{opt}\left[X\right]{\left|X\right|}^{2}\;\;=2\pi N_{0}\left(P\right)\;\;\int_{0}^{\infty }\;\;\frac{d\rho \;\rho^{3}e^{-\lambda_{0}\left(P\right)\rho^{2}}}{\sqrt{1+\gamma^{2}L^{2}\rho^{4}/3}}=P. \end{array}$$

Note that in the leading order in the parameter 1/SNR the function $P_{X}^{opt}\left[X\right]$ depends only on |X|. In the next-to-leading the order in 1/SNR this property holds true as well [20].

In the parametric form the power dependance of the parameters $\lambda_{0}$ and $N_{0}$ reads
(77)$$\begin{array}{cc} & \lambda_{0}\left(P\right)=\frac{\gamma L}{\sqrt{3}\;}\;\alpha ,\;N_{0}\left(P\right)=\frac{\gamma L}{\pi \sqrt{3}\;G\left(\alpha \right)}, \end{array}$$
where $G\left(\alpha \right)=\frac{\pi }{2}\left(H_{0}\left(\alpha \right)-Y_{0}\left(\alpha \right)\right)$, $Y_{0}\left(\alpha \right)$ and $H_{0}\left(\alpha \right)$ are Neumann and Struve functions of zero order, respectively. The parameter α(P) is the real solution of the equation
$$\frac{d}{d\alpha }\log G\left(\alpha \right)=-\gamma LP/\sqrt{3}.$$

Note that the optimal input signal distribution $P_{X}^{opt}\left[X\right]$ (Equation 74) differs from the half-Gaussian distribution (Equation 64).

For sufficiently large values of the power *P*, log(γPL)≫1, we use the asymptotics of $Y_{0}\left(\alpha \right)$ and $H_{0}\left(\alpha \right)$ at small α and arrive at the following result for $\lambda_{0}\left(P\right)$ and $N_{0}\left(P\right)$:(78)$$\begin{array}{c} \lambda_{0}\left(P\right)\approx \frac{1-\log\log\left(B\tilde{\gamma }\right)/\log\left(B\tilde{\gamma }\right)}{P\log(B\tilde{\gamma })},\;N_{0}\left(P\right)\approx \frac{\tilde{\gamma }}{\pi P}\log^{-1}\left[B\tilde{\gamma }/\left(P\lambda_{0}\left(P\right)\right)\right]. \end{array}$$

Here and below we use the notation
(79)$$\begin{array}{c} B=2e^{-\gamma_{E}}, \end{array}$$
(80)$$\begin{array}{c} \tilde{\gamma }=\gamma LP/\sqrt{3} \end{array}$$
is the convenient dimensionless nonlinear parameter. At small *P*, such that the nonlinearity parameter $\tilde{\gamma }\ll 1$, the solutions of the Equations (Equation 75) and (Equation 76) have the form:(81)$$\begin{array}{c} \lambda_{0}\left(P\right)\approx \frac{1-2{\tilde{\gamma }}^{2}}{P},\;N_{0}\left(P\right)\approx \frac{1-{\tilde{\gamma }}^{2}}{\pi P}. \end{array}$$

Note that at $\tilde{\gamma }\to 0$ the optimal distribution (Equation 74) passes to the Gaussian distribution (Equation 65). It is known that this distribution is optimal for a linear channel [1]. In Ref. [20] using the same method we found the first correction to $P_{X}^{opt}\left[X\right]$ proportional to QL.

The fourth, to calculate the mutual information we substitute the expression (Equation 74) for $P_{X}^{opt}\left[X\right]$ in Equations (Equation 71) and (Equation 72) and using the definition (Equation 24) we obtain
(82)$$\begin{array}{c} I_{P_{X}^{opt}\left[X\right]}=C_{0}=P\lambda_{0}\left(P\right)-\log N_{0}\left(P\right)-\log\left(\pi eQL\right). \end{array}$$

The last equation gives the mutual information $I_{P_{X}^{opt}\left[X\right]}$ with the accuracy O(QL).

At small $\tilde{\gamma }$ we obtain
(83)$$\begin{array}{c} I_{P_{X}^{opt}\left[X\right]}\approx \log\left(\mathrm{SNR}\right)-{\tilde{\gamma }}^{2}. \end{array}$$

This result is the Shannon capacity $\log\left(1+\mathrm{SNR}\right)$ at large SNR of the linear channel (Equation 1) with the first nonlinear correction.

In the high power sub-interval ${\left(\gamma L\right)}^{-1}\ll P\ll {\left(QL^{3}\gamma^{2}\right)}^{-1}$ using Equation (Equation 78) one can obtain the following asymptotics for the mutual information in the case of very large nonlinearity parameter ($\log(\tilde{\gamma })\gg 1$):(84)$$\begin{array}{c} I_{P_{X}^{opt}\left[X\right]}=\log\log\left(B\gamma LP/\sqrt{3}\right)-\log\left(QL^{2}\gamma e/\sqrt{3}\right)+\\ \frac{1}{\log\left(B\gamma LP/\sqrt{3}\right)}\left[\log\log\left(B\gamma LP/\sqrt{3}\right)+1-\frac{\log\log\left(B\gamma LP/\sqrt{3}\right)}{\log\left(B\gamma LP/\sqrt{3}\right)}\right]. \end{array}$$

This expression is obtained with the accuracy $1/\log^{2}\left(\tilde{\gamma }\right)$. One can see that the first term on right-hand side of the Equation (Equation 84) grows as loglogP. It means that the mutual information $I_{P_{X}^{opt}\left[X\right]}$ also grows as loglogP at large enough *P*.

Note that the results (Equation 74), (Equation 82), and asymptotics (Equation 78), (Equation 81), (Equation 83), (Equation 84) are obtained in the leading and next-to-leading order in the parameter $1/\sqrt{\mathrm{SNR}}$. Therefore, the results (Equation 74), (Equation 82) are calculated with the accuracy $O\left(1/\mathrm{SNR}\right)$. However, in the literature the bounds of the capacity rather than the asymptotic estimates are on the carpet. Therefore, to find the lower bound of the channel capacity for the per-sample model it is necessary to calculate the corrections of the order of 1/SNR and define their signs. Moreover, to find the applicability region of the result (Equation 82) we have to know the corrections of the order of 1/SNR as well. The applicability region will be defined by the condition that the corrections of the order of 1/SNR are much less than obtained results (Equation 74), (Equation 82). In Ref. [20] we have calculated these corrections using the approach described in Section 3.1. The calculation of these corrections is straightforward but cumbersome, therefore, here we present only the idea of the calculation and demonstrate the results for these corrections for the mutual information. The detailed calculation can be found in Ref. [20].

To calculate the correction to the mutual information we should know the corrections to the conditional probability density function (Equation 53). Therefore, we should calculate the corrections both to the action *S* and to the normalization factor Λ in Equation (Equation 34). To find these corrections we have to calculate the function ϰ(z), see Equation (Equation 51), in the leading, next-to-leading, and next-to-next-to-leading orders in the parameter $1/\sqrt{\mathrm{SNR}}$. Then we should substitute the found corrections for the action *S* to the path integral (Equation 35), and calculate the path integral up to the terms which are proportional to the parameter *Q*. After that, we expand the product of the exponent and the normalization factor Λ up to the terms of order of 1/SNR. This expression is cumbersome, and therefore, we do not present it here, but it can be found in Ref. [20]. To calculate the corrections to the mutual information we substitute the obtained result for P[Y|X] to the Equations (Equation 19)–(Equation 21), (Equation 24). Then, using the method described above, we perform maximization of the mutual information calculated with the accuracy 1/SNR, and obtain the following result for it:(85)$$\begin{array}{c} I_{P_{X}^{opt}\left[X\right]}^{\left(1\right)}=C_{0}+\Delta C\;, \end{array}$$
where $C_{0}$ is defined in (Equation 82). The correction ΔC has the form:(86)$$\begin{array}{c} \Delta C=\frac{1}{\mathrm{SNR}}\left\{\pi N_{0}P\left[\frac{214}{375}-\frac{8}{375}{\left(\frac{\lambda_{0}P}{\tilde{\gamma }}\right)}^{2}\right]+\lambda_{0}P\left[\frac{137}{150}+\frac{8}{375}{\left(\frac{\lambda_{0}P}{\tilde{\gamma }}\right)}^{2}\right]-\frac{347}{750}{\left(\lambda_{0}P\right)}^{2}\right\}. \end{array}$$

The quantity ΔC corresponds to the first non-vanishing correction to the mutual information. One can verify that for the small parameter $\gamma L^{2}Q\ll 1$, the correction (Equation 86) is always small with respect to $C_{0}$. Indeed, the ratio of the expression in the curly brackets in Equation (Equation 86) and $\tilde{\gamma }$ is the bounded function for all values of $\tilde{\gamma }$.

We do not have the explicit analytical result for the correction ΔC for the arbitrary parameter $\tilde{\gamma }$, since the quantities $\lambda_{0}$ and $N_{0}$ are the solutions of the nonlinear equations. But the correction ΔC can be calculated numerically for any parameter $\tilde{\gamma }$. However, for the small and large parameters $\tilde{\gamma }$ we can calculate the asymptotics of the quantity ΔC analytically.

For the small nonlinearity $\tilde{\gamma }\ll 1$ we substitute the parameters $\lambda_{0}$ and $N_{0}$ in the form (Equation 81) to the Equation (Equation 86) and obtain:(87)$$\begin{array}{c} \Delta C\;\approx \frac{1}{\mathrm{SNR}}-\frac{1}{\mathrm{SNR}}\frac{{\tilde{\gamma }}^{2}}{3}. \end{array}$$

Using this result and the asymptotics (Equation 83) we obtain the mutual information within our accuracy in the form:(88)$$\begin{array}{c} I_{P_{X}^{opt}\left[X\right]}^{\left(1\right)}\approx \log\left(1+\mathrm{SNR}\right)-{\tilde{\gamma }}^{2}-\frac{1}{\mathrm{SNR}}\frac{{\tilde{\gamma }}^{2}}{3}. \end{array}$$

One can see that the first term in the right-hand side is the the capacity of the linear channel, the second and the third ones correspond to the nonlinear corrections. The nonlinear corrections in (Equation 88) are negative and they decrease the mutual information $I_{P_{X}^{opt}\left[X\right]}^{\left(1\right)}$ of the channel.

Let us consider the behavior of the correction ΔC at large power *P*. For the case log(γLP)≫1 (but $P\ll {\left(\gamma^{2}QL^{3}\right)}^{-1}$ to be within the intermediate power range) we obtain the simple result:(89)$$\begin{array}{cc} & \Delta C\;\approx \frac{1}{\mathrm{SNR}}\frac{214}{375}\pi N_{0}P. \end{array}$$

Using the asymptotic (Equation 78) for quantity $N_{0}$ we arrive at the expression
(90)$$\begin{array}{ccc} \Delta C\; & \approx & \frac{\gamma L^{2}Q}{\sqrt{3}}\frac{214}{375}{\left(\log\left(B\tilde{\gamma }\log\left(B\tilde{\gamma }\right)\right)+\frac{\log\log(B\tilde{\gamma })}{\log(B\tilde{\gamma })}\right)}^{-1}. \end{array}$$

Note that this correction is suppressed parametrically as $\gamma L^{2}Q$ instead of 1/SNR=QL/P. One can see the correction decreases at large $\tilde{\gamma }$ as $1/\log\tilde{\gamma }$. It is interesting that at large $\tilde{\gamma }$ the correction ΔC is positive, therefore, it slightly enhances the mutual information $I_{P_{X}^{opt}\left[X\right]}^{\left(1\right)}$. Since the correction ΔC is positive in the region defined as log(γLP)≫1, $P\ll {\left(\gamma^{2}QL^{3}\right)}^{-1}$, and the next-to-leading corrections to the mutual information are suppressed parametrically, the quantity $C_{0}$ is the lower bound of the per-sample channel capacity:(91)$$\begin{array}{c} C\geq C_{0}. \end{array}$$

Since there are no corrections of order of $\gamma^{2}L^{3}QP$ at large *P*, see Equation (Equation 90), we expect the next correction containing the power *P* to be of order of ${\left(\gamma^{2}L^{3}QP\right)}^{2}$, see Ref. [19]. Therefore, the applicability region at large *P* for the quantity $C_{0}$ is determined by the condition ${\left(\gamma^{2}L^{3}QP\right)}^{2}\ll 1$. For the given small parameter $\gamma^{2}L^{3}QP$ this condition extends the applicability region for the lower bound of the channel capacity $C_{0}$. For realistic channel parameters presented in the Table 1.

We have the following intermediate power range:(92)$$\begin{array}{c} 1.5\times 10^{-4}\mathrm{mW}\ll P\ll 0.66\times 10^{4}\mathrm{mW}. \end{array}$$

One can see that the range is very wide. For the presented parameters we have numerically calculated the lower bound $C_{0}$ using Equations (Equation 77) and (Equation 82). The result of the calculation is presented in Figure 2. Also in Figure 2 we present the comparison of the approximation (Equation 85) with the Shannon capacity of a linear channel and with the asymptotic capacity bound (Equation 70).

One can see that the Shannon capacity of the linear channel with the additive noise (the red dashed-dotted line in the Figure 2) is always greater than the lower bound (Equation 82) for the nondispersive nonlinear fiber channel (the black solid line in the Figure 2) for the considered range of *P*. But the lower bound (Equation 82) is greater than the asymptotic capacity bound (Equation 70) in the intermediate power range.

In Ref. [22] the comparison of our result (Equation 82) for the NLSE per-sample model (authors referred to our model as the memoryless NLS channel (MNC)) with two other models, more precisely, the regular perturbative channel (RPC) and the logarithmic perturbative channel (LPC), was performed. The comparison was illustrated in Figure 1 of Ref. [22], and we present this figure in the Figure 3.

The authors claimed that they established a novel upper bound on the capacity of the NLSE per-sample model (the violet curve $U_{MNC}\left(P\right)$ in Figure 3). In addition, the authors considered various input signal distributions within MNC: half-Gaussian (Equation 64), Gaussian (Equation 65), and modified Gaussian distribution (Equation 63) optimized in parameter β (it is denoted as “Max-chi” in Figure 3). They used the following channel parameters: $\gamma =1.27\;W^{-1}{\mathrm{km}}^{-1}$, L=5000km, $QL=7.36\times 10^{-3}\mathrm{mW}$. For these parameters the upper limit of the intermediate power range (we choose it as $P_{max}=6\pi^{2}{\left(Q\gamma^{2}L^{3}\right)}^{-1}$, see [18]) is estimated as $P_{max}=0.2\;W=23\;\mathrm{dBm}$. It is obvious in Figure 3, that up to this power $P_{max}$ our result (green solid line) is consistent with the capacity upper bound of the per-sample model (the violet curve $U_{MNC}$) and it exceeds other mutual information curves for other input signal distributions in all intermediate power range. Of course, for large powers ($P≳P_{max}$) our input signal distribution (Equation 74) is not the optimal any more, and the mutual information (Equation 85) underestimates the real capacity. In the strict sense, the corrections to (Equation 85) are small only up to P∼5.5dBm, and when P>5.5dBm we have the transition range from the intermediate power range with the optimal input signal distribution (Equation 74) to the large power semirange where the optimal distribution is believed to be the half-Gaussian one, see [17,18]. The large extension of the transition range (up to $P_{max}∼23\;\mathrm{dBm}$) can be explained by the smallness of the next-to-leading order corrections (Equation 86) and its decreasing when increasing the power, see Equation (Equation 90).

To finalize the per-sample channel consideration we emphasize the main results. We developed the path integral approach to the calculation of the conditional PDF P[Y|X] for large SNR. We demonstrated that for the nonlinear nondispersive channel the lower bound $C_{0}$ of the capacity increases only as loglogP at large signal power *P* instead of the behavior logP that is specific for the channel with zero nonlinearity. To determine the applicability region and the accuracy of the found quantity $C_{0}$ we calculated the first non-zero correction ΔC proportional to the noise power QL in the intermediate power range $QL\ll P\ll {\left(\gamma^{2}L^{3}Q\right)}^{-1}$. We demonstrated that the quantity ΔC is small in the intermediate power range, and it is the positive decreasing function at large signal power *P*: ${\left(\gamma L\right)}^{-1}\ll P\ll {\left(\gamma^{2}L^{3}Q\right)}^{-1}$. The found result is in agreement with recent results of Ref. [22].

### 3.2. Extended Model: Considerations of the Time Dependent Input Signals

Let us start this section from the consideration of the conditional probability density function P[Y|X]. In the case of the nontrivial time dependance of the input X(t) and output Y(t) signals the conditional PDF reads Ref. [11]:(93)$$\begin{array}{cc} & P\left[Y\left(t\right)|X\left(t\right)\right]=\;\;\;\;\int_{\psi (0,t)=X(t)}^{\psi (L,t)=Y(t)}\;\;\;\;D\psi \;e^{-S[\psi ]/Q}\;, \end{array}$$
where the effective action S[ψ] has the form: $S\left[\psi \right]=\int_{0}^{L}dz\int dt|\partial_{z}\psi \left(z,t\right)-i\gamma {\left|\psi \left(z,t\right)\right|}^{2}\psi \left(z,t\right){|}^{2}$; the integration measure Dψ(z,t) depends on both z− and t− discretization scheme:(94)$$\begin{array}{c} D\psi (z,t)={\left(\frac{\Delta }{\Delta_{z}\pi Q}\right)}^{2M}\prod_{j=-M}^{M-1}\prod_{i=1}^{N_{z}-1}\left\{\frac{\Delta }{\Delta_{z}\pi Q}\;\;dRe\psi \left(z_{i},t_{j}\right)\;dIm\psi \left(z_{i},t_{j}\right)\right\}, \end{array}$$
where $\Delta =T/\left(2M\right)=\left(2N+1\right)T_{0}/\left(2M\right)$ is the time grid spacing and $\Delta_{z}=L/N_{z}$ is the z-coordinate grid spacing. Of course, the expression (Equation 93) contains all information about the transmitted signal and its interaction with the noise, but as strange as it sounds, the expression (Equation 93) contains redundant information (i.e., degrees of freedom which cannot be detected). The point is that in the realistic communication channel the receiver has the finite bandwidth, it means that it reduces somehow the bandwidth of the received signal Y(t). After that, there is a procedure of the extraction of the information from the signal measured by receiver. This detection procedure should be implemented in the function P[Y(t)|X(t)].

To demonstrate the point, let us consider the input signal X(t) in the form (Equation 11). In such form, the number of the degrees of freedom in the path integral (Equation 93) is infinite, if the function X(t) is continuous, or 2M degrees of freedom if we set the function X(t) in the discrete-time moments $t_{i}$, see the text after Equation (Equation 17). However, the transmitted information is carried only by 2N+1 coefficients $C_{n}$ (N≪M), see Equation (Equation 11). It means that to obtain the conditional probability density function which describes the transmission of the information carried by the set of the coefficients $C_{n}$ we have to integrate over the redundant degrees of freedom. The approach based on the path integral representation (Equation 93) is general and it can be used for any signal model, any receiver and projecting procedure (Equation 18). For our signal model described in the Section 2.2, we can use a simpler method of the calculation of the conditional PDF $P[\left\{\tilde{C}\right\}|\left\{C\right\}]$. Below we describe the method.

In our model, the signal propagation for different time moments $t_{j}$ is independent because the dispersion is zero and the noise is not correlated for different time moments $t_{i}\neq t_{j}$. Therefore, the conditional PDF P[Y(t)|X(t)] can be presented in the factorized form:(95)$$\begin{array}{c} P\left[Y\left(t\right)|X\left(t\right)\right]=\prod_{j=-M}^{M-1}P_{j}\left[Y_{j}|X_{j}\right], \end{array}$$
where $X_{j}=X\left(t_{j}\right)$, $Y_{j}=Y\left(t_{j}\right)$, and $P_{j}\left[Y_{j}|X_{j}\right]$ is per-sample conditional PDF described in the previous Section 3.1.1, where we should replace $Y\to Y_{j}$, $X\to X_{j}$, Q→Q/Δ.

Our goal is to find the PDF $P[\left\{\tilde{C}\right\}|\left\{C\right\}]$ in the leading order in the parameter 1/SNR. Instead of calculation of the path integral, we build the PDF which reproduces all possible correlators of ${\tilde{C}}_{k}$: $\langle {\tilde{C}}_{k_{1}}\rangle$, $\langle {\tilde{C}}_{k_{1}}{\tilde{C}}_{k_{2}}\rangle$, $\langle {\tilde{C}}_{k_{1}}\ldots {\bar{\tilde{C}}}_{k_{n}}\rangle$ for the fixed input set {C} in the leading order in the parameter Q. These correlators read as
(96)$$\begin{array}{c} \langle {\tilde{C}}_{k_{1}}\ldots {\bar{\tilde{C}}}_{k_{n}}\rangle =\int \prod_{j=-M}^{M-1}d^{2}Y_{j}P\left[Y\left(t\right)|X\left(t\right)\right]{\tilde{C}}_{k_{1}}\ldots {\bar{\tilde{C}}}_{k_{n}}, \end{array}$$
where d2Yj=dReYjdImYj, and ${\tilde{C}}_{k}$ is defined in Equation (Equation 18). After substitution of Equations (Equation 95), (Equation 53), and (Equation 18) into Equation (Equation 96) and performing the integration we obtain in the leading order in the noise parameter Q:(97)$$\begin{array}{cc} & \langle {\tilde{C}}_{k}\rangle =C_{k}-\frac{iC_{k}\;QL^{2}\gamma }{\Delta }\left(1-\frac{i\gamma L|C_{k}{|}^{2}n_{4}}{3}\right), \end{array}$$
(98)$$\begin{array}{c} \langle \left({\tilde{C}}_{m}-\langle {\tilde{C}}_{m}\rangle \right)\left({\tilde{C}}_{n}-\langle {\tilde{C}}_{n}\rangle \right)\rangle =-i\delta_{m,n}\frac{C_{m}^{2}QL^{2}\gamma }{T_{0}}\left(n_{4}-\frac{2in_{6}}{3}\gamma L{\left|C_{m}\right|}^{2}\right), \end{array}$$
(99)$$\begin{array}{cc} & \langle \left({\tilde{C}}_{m}-\langle {\tilde{C}}_{m}\rangle \right)\bar{\left({\tilde{C}}_{n}-\langle {\tilde{C}}_{n}\rangle \right)}\rangle =\delta_{m,n}\frac{QL}{T_{0}}\left(1+\frac{2n_{6}}{3}\gamma^{2}L^{2}{\left|C_{m}\right|}^{4}\right), \end{array}$$
where $\delta_{m,n}$ is the Kronecker symbol and we have introduced the following notation for the integral of the *s*-th power of the pulse envelope function f(t):(100)$$\begin{array}{c} n_{s}=\int_{-T_{0}/2}^{T_{0}/2}\frac{dt}{T_{0}}f^{s}\left(t\right). \end{array}$$

We remind that f(t) is assumed to be normalized by the condition $n_{2}=1$. Note that the correlator $\langle {\tilde{C}}_{k}-C_{k}\rangle$ is proportional to $QL/\Delta =QLW^{\prime }/\left(2\pi \right)$, i.e., it is proportional to the total noise power in the whole bandwidth $W^{\prime }$. The reason for that is the bandwidth of the receiver coincides with the bandwidth of the noise. The correlators (Equation 98) and (Equation 99) are proportional to $QL/T_{0}$ and do not depend on the discretization parameter Δ. It means that these correlators depend only on the bandwidth of the envelope function f(t). So we obtain that in the leading order in the parameter Q the shift of the mean value ${\tilde{C}}_{k}$ due to the signal-noise interaction is proportional to the total noise power in the channel ($W^{\prime }=2\pi /\Delta$), whereas the spread around the average value, see (Equation 98), (Equation 99), is proportional to the noise power containing in the bandwidth *W* of the pulse envelope (bandwidth of the pulse envelope coincides with the bandwidth of the signal X(t)). Note that the higher-order corrections in parameter Q to the correlators are more complicated and contain the noise bandwidth, see details in Appendix A of Ref. [21]. The correlators of higher orders in $\tilde{C}$ can be calculated in the leading order in the parameter Q using Equations (Equation 97)–(Equation 99).

To verify the analytical results (Equation 97)–(Equation 99) the numerical simulations of pulse propagation through a nonlinear nondispersive optical fiber were performed in Ref. [21]. After that the numerical results for correlators (Equation 97)–(Equation 99) were obtained. To find these correlators the Equation (Equation 4) was solved numerically for the fixed input signal X(t) and for different realizations of the noise η(z,t). After that the detection procedure described by Equations (Equation 17) and (Equation 18) was applied. Finally, the averaging procedure over noise realizations for the coefficients ${\tilde{C}}_{k}$ was performed. Two numerical methods of the solution of Equation (Equation 4) were used: the split-step Fourier method and the Runge–Kutta method of the fourth order. It was shown that the numerical results do not depend on the numerical method and these results are in consistent with analytical ones for different realizations of the input pulse envelope f(t) and the noise bandwidth $W^{\prime }$. Below we present the comparison of the numerical and analytical results obtained in Ref. [21]. The numerical simulation was done for the following channel parameters, see the Table 2.

We choose the duration of one pulse as $T_{0}=10^{-10}$ s. Simulations were performed for different *t*-meshes, i.e., for different time grid spacing Δ, and for different pulse envelopes. Different grid spacings Δ correspond to the different noise bandwidths $W^{\prime }=2\pi /\Delta$ for the fixed noise parameter Q. The numerical calculations were performed for the different Δ presented in the Table 3.

The different grid spacings determine the different widths of the conjugated ω-meshes in the frequency domain: $1/\Delta_{1}=10.26$ THz, $1/\Delta_{2}=5.12$ THz and $1/\Delta_{3}=2.56$ THz. In Ref. [21] the different envelopes f(t) were considered as well. Here we present results only for the Gaussian envelope:(101)$$f\left(t\right)=\sqrt{\frac{T_{0}}{T_{1}\sqrt{\pi }}}\exp\left(-\frac{t^{2}}{2T_{1}^{2}}\right),$$
where $T_{1}=T_{0}/10=10^{-11}$ s stands for the characteristic time scale of the function f(t). Such relation between $T_{0}$ and $T_{1}$ means that the overlapping between different pulses is negligible. For pulses with envelope (Equation 101) the coefficients $n_{s}$ defined in Equation (Equation 100) are $n_{4}=\frac{T_{0}/T_{1}}{\sqrt{2\pi }}\approx 3.989$, $n_{6}=\frac{{\left(T_{0}/T_{1}\right)}^{2}}{\pi \sqrt{3}}\approx 18.38$, $n_{8}=\frac{{\left(T_{0}/T_{1}\right)}^{3}}{2\pi \sqrt{\pi }}\approx 89.79$. In Figure 4 and Figure 5 the real and imaginary parts of the quantity $\left(C_{k}-\langle {\tilde{C}}_{k}\rangle \right)/C_{k}$ as a function of $|C_{k}{|}^{2}$ are depicted for different values of grid spacing Δ.

One can see the good agreement between analytical and numerical results depicted in Figure 4. There is some difference in the imaginary part of analytical and numerical results corresponding to grid spacing $\Delta_{1}$ at large $|C_{k}|$, see Figure 5. The reason is that, for the analytical results corresponding to $\Delta_{1}$ it is necessary to take into account the next corrections in the parameter Q Ref. [21]. The numerical and analytical results for the correlators (Equation 98), (Equation 99) are also in a good agreement, for details see Ref. [21]. In this paper, it was demonstrated that the relative importance of the next-to-leading order corrections for the correlators (Equation 98) and (Equation 99) is governed by the dimensionless parameter ${\left(\frac{QL}{\Delta }\gamma L\right)}^{2}\gamma LP$, i.e., it increases linearly for increasing power *P*.

Now we can proceed to the search for the conditional probability density function. Using the correlators (Equation 97)–(Equation 99) we build the conditional PDF $P[\left\{\tilde{C}\right\}|\left\{C\right\}]$ which reproduces all correlators of the coefficients ${\tilde{C}}_{m}$ in the leading order in parameter Q. Thus, the conditional PDF has the form [21]:(102)$$\begin{array}{c} P\left[\left\{\tilde{C}\right\}|\left\{C\right\}\right]=\prod_{m=-N}^{N}P_{m}\left[{\tilde{C}}_{m}|C_{m}\right], \end{array}$$
where
(103)$$\begin{array}{c} P_{m}\left[{\tilde{C}}_{m}|C_{m}\right]\approx T_{0}\frac{\exp\left[-T_{0}\frac{\left(1+4n_{6}\mu_{m}^{2}/3\right)x_{m}^{2}+2x_{m}y_{m}\mu_{m}n_{4}+y_{m}^{2}}{QL\left(1+\xi^{2}\mu_{m}^{2}/3\right)}\right]}{\pi QL\sqrt{1+\xi^{2}\mu_{m}^{2}/3}}, \end{array}$$
(104)xm=Ree−iϕmC˜m−Cm+iCmQL2γΔ1−iγL|Cm|2n43,
(105)ym=Ime−iϕmC˜m−Cm+iCmQL2γΔ1−iγL|Cm|2n43,
where $\phi_{m}=\arg C_{m}$, $\mu_{m}=\gamma L{\left|C_{m}\right|}^{2}$, and
(106)$$\begin{array}{c} \xi^{2}=\left(4n_{6}-3n_{4}^{2}\right). \end{array}$$

The parameter $\xi^{2}$ obeys inequality $\xi^{2}\geq n_{6}>0$ due to Cauchy–Schwartz–Buniakowski inequality. For the Gaussian envelope (Equation 101) this parameter is
(107)$$\begin{array}{c} \xi^{2}=\frac{1}{\pi }{\left(\frac{T_{0}}{T_{1}}\right)}^{2}\left(\frac{4}{\sqrt{3}}-\frac{3}{2}\right)\approx 0.258{\left(\frac{T_{0}}{T_{1}}\right)}^{2}, \end{array}$$
and for the chosen parameters $T_{1}=T_{0}/10=10^{-11}$ sec one obtains ξ≈5.08.

Equation (Equation 102) means that our channel decomposes to the 2N+1 independent information channels. Therefore, the function $P_{m}\left[{\tilde{C}}_{m}|C_{m}\right]$ describes the channel corresponding to the *m*-th time slot. The function $P_{m}\left[{\tilde{C}}_{m}|C_{m}\right]$ obeys the normalization condition
(108)$$\begin{array}{c} \int d^{2}{\tilde{C}}_{m}P_{m}\left[{\tilde{C}}_{m}|C_{m}\right]=1. \end{array}$$

Since there are 2N+1 independent channels, we can choose the input signal distribution $P_{X}\left[\left\{C_{m}\right\}\right]$ in the factorized form:(109)$$\begin{array}{c} P_{X}\left[\left\{C\right\}\right]=\prod_{k=-N}^{N}P_{X}^{\left(k\right)}\left[C_{k}\right], \end{array}$$
and we can consider only one channel, say *m*-th channel. One can see that the presentation of the conditional PDF (Equation 103) is close to the presentation (Equation 53) for the per-sample PDF. Also the function $P_{m}\left[{\tilde{C}}_{m}|C_{m}\right]$ changes significantly when the variable $C_{m}$ changes on the value of order of $\sqrt{QL/T_{0}}$ for fixed value of ${\tilde{C}}_{m}$. Such behavior coincides with that for the function P[Y|X] for the per-sample model. Below, we imply that
(110)$$\begin{array}{c} P\gg QL/\Delta \gg QL/T_{0}, \end{array}$$
where *P* is the mean power of the *m*-th pulse, i.e., the signal power is much greater than the noise power in the whole bandwidth $W^{\prime }=2\pi /\Delta$ and in the input signal bandwidth *W*. For the extended model we define the signal-to-noise ratio as
(111)$$\begin{array}{c} \mathrm{SNR}=\frac{PT_{0}}{QL}, \end{array}$$
on the assumption (Equation 110) we have SNR≫1. We also imply that the PDF of the input signal $P_{X}^{\left(m\right)}\left[C_{m}\right]$ is a smooth function that changes on a scale $|C_{m}|∼\sqrt{P}$. Therefore, using a consideration similar to that for the per-sample model we obtain that the PDF of the output signal reads
(112)$$\begin{array}{c} P_{out}^{\left(m\right)}\left[{\tilde{C}}_{m}\right]=\int d^{2}C_{m}P_{m}\left[{\tilde{C}}_{m}|C_{m}\right]P_{X}^{\left(m\right)}\left[C_{m}\right] \end{array}$$
in the leading order in the parameter 1/SNR it has the form:(113)$$\begin{array}{c} P_{out}^{\left(m\right)}\left[{\tilde{C}}_{m}\right]\approx P_{X}^{\left(m\right)}\left[{\tilde{C}}_{m}\right]. \end{array}$$

Just as a reminder, to obtain result (Equation 113) we perform the integration in Equation (Equation 112) using Laplace’s method [27], see details in Ref. [19].

Since we know the conditional PDF $P_{m}\left[{\tilde{C}}_{m}|C_{m}\right]$ and output signal PDF $P_{out}^{\left(m\right)}\left[{\tilde{C}}_{m}\right]$ the entropies $H[{\tilde{C}}_{m}]$, $H[{\tilde{C}}_{m}|C_{m}]$ and then mutual information $I_{P_{X}^{\left(m\right)}}$ can be calculated. The calculation is similar to that performed in the Section 3.1.3. After these calculations we solve the variational problem for the mutual information $I_{P_{X}^{\left(m\right)}}$, and find the optimal input signal PDF in the form:(114)$$\begin{array}{c} P_{opt}^{\left(m\right)}\left[C_{m}\right]=N_{0}\frac{e^{-\lambda |C_{m}{|}^{2}}}{\sqrt{1+\xi^{2}\gamma^{2}L^{2}{\left|C_{m}\right|}^{4}/3}}. \end{array}$$

Substituting the result (Equation 114) to the mutual information, we obtain the following result:(115)$$\begin{array}{c} I_{P_{opt}^{\left(m\right)}}=\log\left(\frac{PT_{0}}{\pi eQL}\right)+P\lambda -\log\left[PN_{0}\right], \end{array}$$
where the parameters $N_{0}$ and λ are the solutions (as functions of power *P*) of the normalization conditions for the function (Equation 114):(116)$$\begin{array}{c} \int_{0}^{\infty }d\rho \frac{2\pi N_{0}\;\rho \;e^{-\lambda \rho^{2}}}{\sqrt{1+\xi^{2}\gamma^{2}L^{2}\rho^{4}/3}}=1,\;\int_{0}^{\infty }\;\;d\rho \frac{2\pi N_{0}\;\rho^{3}e^{-\lambda \rho^{2}}}{\sqrt{1+\xi^{2}\gamma^{2}L^{2}\rho^{4}/3}}=P, \end{array}$$
where we have performed the change of variables, $|C_{m}|$ to ρ. Note that the results (Equation 115)–(Equation 116) are obtained in the leading order in the parameter 1/SNR. Note that the expression (Equation 115) is obtained for one *m*-th pulse, therefore, to obtain the mutual information of the channel with the input signal (Equation 11) it is necessary to multiply the right-hand-side of Equation (Equation 115) by the number of the independent channels, i.e., 2N+1.

One can see that relations in the Equation (Equation 116) coincide with ones in the Equation (Equation 76) after changing the parameter γ→γ/ξ. Therefore, to obtain the results for the mutual information $I_{P_{opt}^{\left(m\right)}}$ and its asymptotics for the extended model we can replace the parameter γ with ξγ and parameter $Q\to Q/T_{0}$ in Equations (Equation 82)–(Equation 84). So, by the rescaling of Figure 2 we obtain the mutual information $I_{P_{opt}^{\left(m\right)}}$ of the extended model, see Figure 6.

The asymptotics has the form:(117)$$\begin{array}{c} I_{P_{opt}^{\left(m\right)}}\approx \log\left(\frac{PT_{0}}{QL}\right)-\frac{\xi^{2}\gamma^{2}L^{2}P^{2}}{3}, \end{array}$$
for ξγLP≪1, and for the case when power *P* obeys the conditions logξγLP≫1, $P\ll T_{0}/\left(QL^{3}\xi^{2}\gamma^{2}\right)$ we have the asymptotics
(118)$$\begin{array}{c} I_{P_{opt}^{\left(m\right)}}\approx \log\log\left(B\xi \gamma LP/\sqrt{3}\right)-\log\left(QL^{2}\xi \gamma e/\left(T_{0}\sqrt{3}\right)\right)+\\ \frac{1}{\log\left(B\xi \gamma LP/\sqrt{3}\right)}\left[\log\log\left(B\xi \gamma LP/\sqrt{3}\right)+1-\frac{\log\log\left(B\xi \gamma LP/\sqrt{3}\right)}{\log\left(B\xi \gamma LP/\sqrt{3}\right)}\right]. \end{array}$$

Note that the asymptotics (Equation 118) is obtained with accuracy $1/\log^{2}\left(\xi \gamma LP\right)$.

The found mutual information (Equation 115) is calculated for the fixed shape of the pulse envelope f(t). It is worth noting that the pulse shape f(t) is encoded only through one parameter ξ, see Equation (Equation 106). Therefore, strictly speaking, to find the capacity one should find the maximum over all forms of the pulses. On the one hand, we can consider the expression (Equation 115) as the estimation of the capacity of the channel whose model implies the specific given shape of the pulse envelope f(t). On the other hand, using Equation (Equation 118) for fixed power *P* one can increase the value of $I_{P_{opt}^{\left(m\right)}}$ raising the parameter $\xi^{2}$, say, decreasing the signal time width $T_{1}$ for the Gaussian pulse envelope, see Equation (Equation 107). However, the limitation of the intermediate power range $P\ll T_{0}/\left(QL^{3}\xi^{2}\gamma^{2}\right)$ makes this parameter have an upper limit $\xi_{\max}^{2}∼{\left[\gamma LP(\gamma L^{2}Q/T_{0})\right]}^{-1}$, and the maximum value of the mutual information $I_{P_{opt}^{\left(m\right)}}$ over pulse envelope parameters is attained for the argument $B\xi_{\max}\gamma LP/\sqrt{3}∼\sqrt{\frac{PT_{0}}{QL}}=\sqrt{\mathrm{SNR}}$.

Note that the mutual information $I_{P_{opt}^{\left(m\right)}}$, see Equation (Equation 115), cannot be considered as the lower bound of the channel capacity since we do not know the sign of the next-to-leading order corrections in the parameter Q. However, the quantity $I_{P_{opt}^{\left(m\right)}}$ is the estimation of the capacity with the accuracy O(Q). The mutual information $I_{P_{opt}^{\left(m\right)}}$ grows as loglogP, see Equation (Equation 118), for sufficiently large average power *P*: ${\left(\xi \gamma L\right)}^{-1}\ll P\ll T_{0}/\left(QL^{3}\xi^{2}\gamma^{2}\right)$. Note that we have obtained similar asymptotics for the per-sample model. The time dependence of the pulse leads to same asymptotics behavior with modified nonlinearity parameter γ (γ→ξγ).

It worth noting that all our calculations were performed under the assumption of the negligible overlapping of the pulses, see Equation (Equation 12). It means we imply that the corrections due to the overlapping effects are at least of the same order as the next-to-leading order corrections in the parameter QL. We can satisfy this condition by choosing appropriate pulse parameters $T_{0}$ and $T_{1}$.

## 4. Conclusions

In our review, we considered nondispersive nonlinear optical-fiber communication channel with the additive noise. We studied two different models of the channel: the per-sample model and the extended model. For these models, we present results of the calculation for the following theoretical information characteristics such as the output signal entropy, conditional entropy, mutual information in the leading order in the parameter 1/SNR. To calculate these quantities two methods were developed for the calculation of the conditional probability density function P[Y|X] [19,21]. The first method which was used for the P[Y|X] calculation for the per-sample model is based on the path integral approach. In the approach, the path integral (Equation 28) was treated using the saddle point method, i.e., the expansion in the parameter 1/SNR, see Refs. [19,20]. This method was used to obtain the expression for the conditional PDF P[Y|X] which is convenient for the analytical calculations of the output signal PDF $P_{out}\left[Y\right]$, entropies, the mutual information, and the optimal input signal distribution in the leading [19], next-to-leading [20] orders in the parameter 1/SNR. These calculations allow us to find the lower bound of the channel capacity for the per-sample model in the intermediate power range. The second method was applied to the investigation of the extended model, which contains such characteristics as the bandwidths of the input signal, the noise, and the receiver, and also takes into account the projection procedure (Equation 18). The second method is based on the calculation of the correlators of the output signal for the fixed input signal in the leading and next-to-leading orders in the noise parameter Q, see Ref. [21]. Using these correlators the conditional PDF $P[\left\{\tilde{C}\right\}|\left\{C\right\}]$ which reproduces all these correlators in the leading order in the parameter 1/SNR was constructed. The knowledge of the function $P[\left\{\tilde{C}\right\}|\left\{C\right\}]$ allowed us to find the informational characteristics of the extended model [21].

We compared the results of our calculations for the per-sample model with limitations on the capacity obtained by other authors [4,17,18,22]. We demonstrated that the conditional PDF P[Y|X] presented even in the leading order in the parameter 1/SNR reproduces the exact result (Equation 29) with high accuracy, see Figure 1. Using the expression (Equation 53) in the intermediate power range, we found the lower bound (Equation 82) of the capacity which is consistent with the recent results obtained in Ref. [22].

For the extended model [21] we present results of the calculation of the output signal correlators for the fixed input signal and demonstrate that the difference in the average value for the recovered coefficient ${\tilde{C}}_{k}$ and the input coefficient $C_{k}$ is proportional to the noise power containing in the total noise bandwidth, see Equation (Equation 97). Whereas the covariances (Equation 98), (Equation 99) are proportional to the noise power containing in the input signal bandwidth. This behavior of the mean value $\langle {\tilde{C}}_{k}\rangle$ is related to the model of the receiver (i.e., the bandwidth of the receiver coincides with the bandwidth of the noise). The obtained analytical results (Equation 97)–(Equation 99) were confirmed by the direct numerical calculations, see Figure 4 and Figure 5. Therefore, the constructed conditional PDF $P[\left\{\tilde{C}\right\}|\left\{C\right\}]$ contains information about the bandwidths of the signal, noise and receiver [21]. This result is in agreement with assertions made in Ref. [14]. Despite the dependance of the PDF $P[\left\{\tilde{C}\right\}|\left\{C\right\}]$ on the noise bandwidth the mutual information calculated in the leading order in 1/SNR for the extended model depends only on the noise power containing in the input signal bandwidth. Since we have not calculated the corrections in the parameter 1/SNR to the mutual information (Equation 115), we can only to consider this quantity as the capacity estimation rather than the lower bound of the capacity.

The models considered in the present paper are widely-spaced from the modern communication systems where the coefficient of the second dispersion is not zero and the signal detection procedure differs from considered above. The effects related to the non-zero dispersion and properties of the receiver can significantly change the results for the mutual information obtained in our consideration. However, the methods described in the present paper may be useful for the consideration of real communication systems. The calculations performed in Refs. [29,30,31,32,33] is indicative of the possibility to use the presented methods for the capacity investigation of the nonlinear fiber-optical channels with non-zero dispersion.

## Figures and Tables

**Figure 1 entropy-22-00607-f001:**
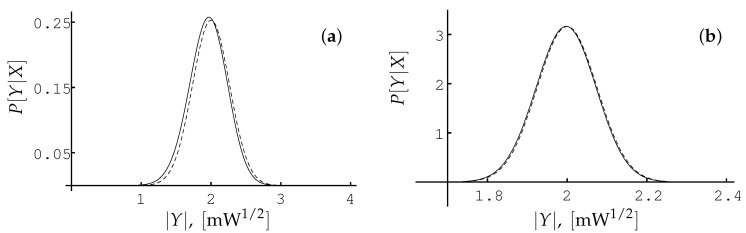
The dependence of the probability density function P[Y|X] on |Y| for $X=2\;{\mathrm{mW}}^{1/2}$ and the argument arg(Y)=μ=4. The plot (**a**) corresponds to signal-to-noise power ratio (SNR) = 8, the plot (**b**) corresponds to SNR=100. The solid line and dashed line correspond to the exact expression (Equation 29) and approximation (Equation 53), correspondingly.

**Figure 2 entropy-22-00607-f002:**
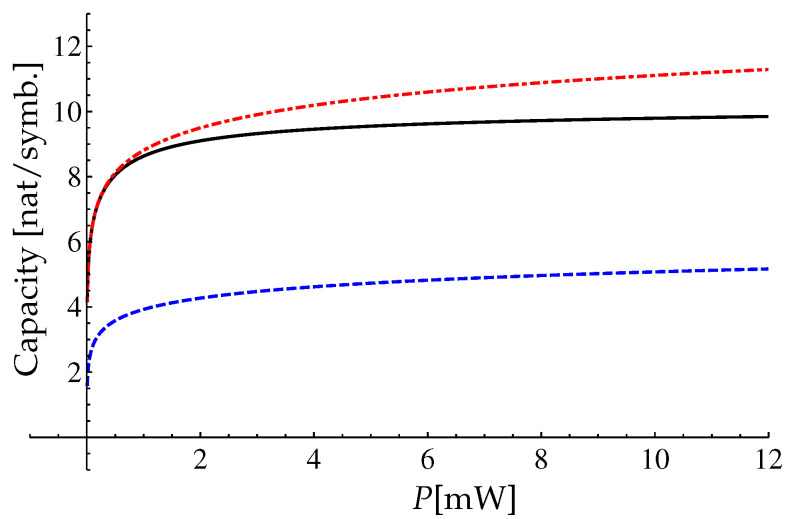
Shannon capacity, the lower bound $C_{0}$, and the asymptotic capacity bound (Equation 70) for the channel parameters from the Table 1. The red dashed-dotted line corresponds to the Shannon limit log(1+SNR), the black solid line corresponds to the lower bound $C_{0}$, see Equation (Equation 82), the blue dashed line corresponds to the bound (Equation 70).

**Figure 3 entropy-22-00607-f003:**
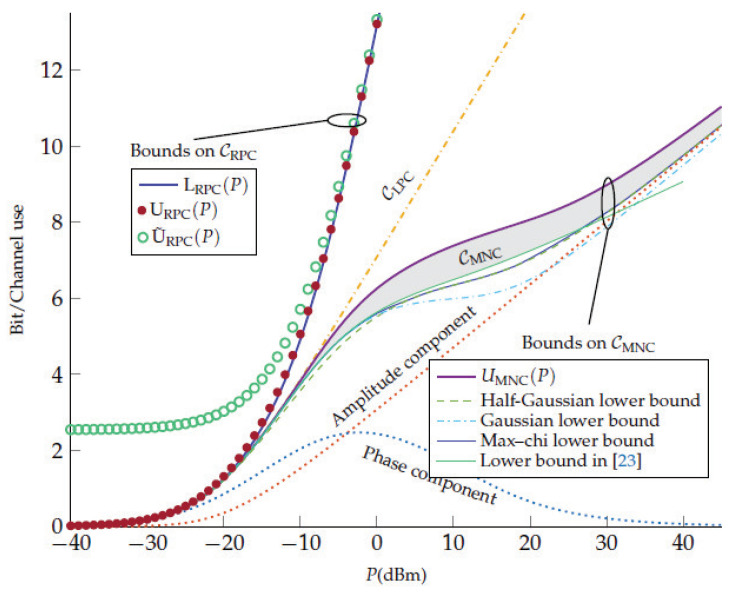
[This figure is taken from Ref. [22]]. Capacity bounds for the RPC and LPC models of Ref. [22], together with the capacity of the per-sample NLSE model. The channel parameters are as follows: $\gamma =1.27\;W^{-1}{\mathrm{km}}^{-1}$, L=5000km, $QL=7.36\times 10^{-3}\mathrm{mW}$. Our result (Equation 82) is presented by the green solid line (“lower bound in [23]”). The intermediate power range goes out from P∼−20dBm to P∼6dBm.

**Figure 4 entropy-22-00607-f004:**
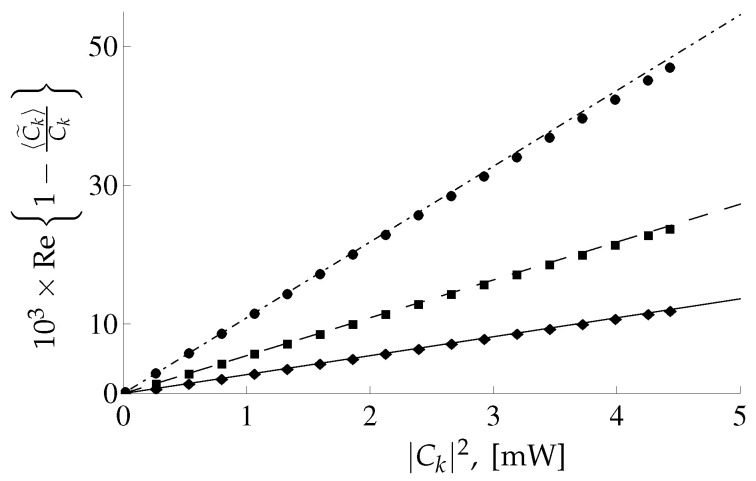
[This figure is taken from Ref. [21]]. The real part of the relative difference of the coefficient $C_{k}$ and the correlator (Equation 97) in units $10^{-3}$ as a function of $|C_{k}{|}^{2}$, see [21]. Dashed-doted, dashed, and solid lines correspond to an analytic representation (Equation 97) for time grid spacings $\Delta_{1}$, $\Delta_{2}$, $\Delta_{3}$, respectively. Circles, squares, and diamonds correspond to numerical results for time grid spacings $\Delta_{1}$, $\Delta_{2}$, $\Delta_{3}$, respectively.

**Figure 5 entropy-22-00607-f005:**
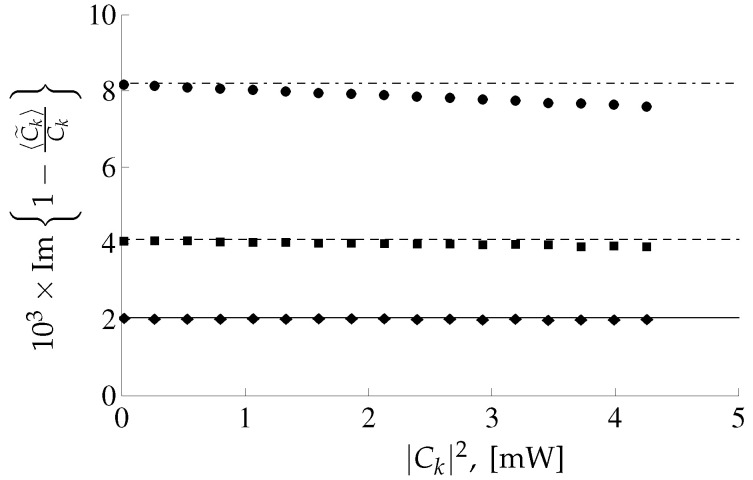
[This figure is taken from Ref. [21]]. The imaginary part of the relative difference of the coefficient $C_{k}$ and the correlator (Equation 97) in units $10^{-3}$ as a function of $|C_{k}{|}^{2}$, see [21]. Dashed doted, dashed, and solid lines correspond to analytic representation (Equation 97) for time grid spacings $\Delta_{1}$, $\Delta_{2}$, $\Delta_{3}$, respectively. Circles, squares, and diamonds correspond to numerical results for time grid spacings $\Delta_{1}$, $\Delta_{2}$, $\Delta_{3}$, respectively.

**Figure 6 entropy-22-00607-f006:**
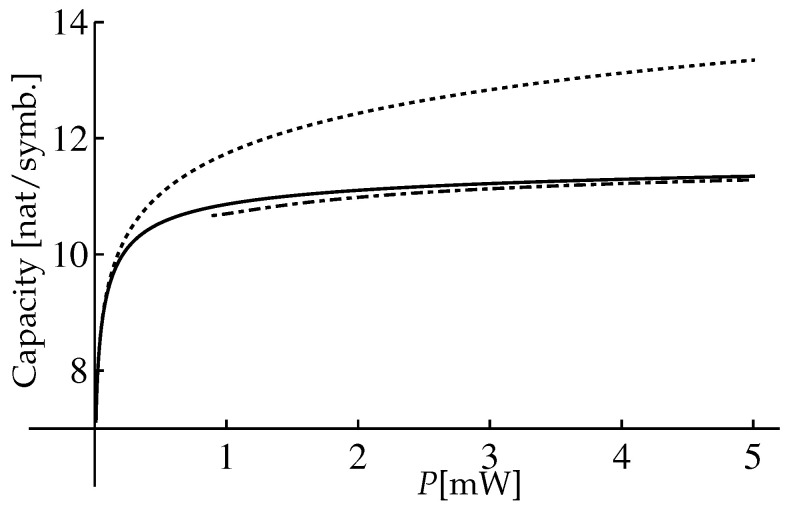
[This figure is taken from Ref. [21]]. Shannon capacity and the mutual information $I_{P_{opt}^{\left(m\right)}}$ for the parameters which are presented in the Table 2, $T_{0}=10^{-10}$ sec, and for the Gaussian shape (Equation 101) of f(t). The black dotted line corresponds to the Shannon limit $\log\left(\frac{PT_{0}}{QL}\right)$, the black solid line corresponds to $I_{P_{opt}^{\left(m\right)}}$, see Equation (Equation 115), the black dashed dotted line corresponds to the asymptotics (Equation 118) for large γLP.

**Table 1 entropy-22-00607-t001:** Channel parameters for the per-sample model.

γ [(km×mW)${}^{-1}$]	*Q* [mW/(km)]	*L* [km]
$10^{-3}$	$1.5\times 10^{-7}$	$10^{3}$

**Table 2 entropy-22-00607-t002:** Channel parameters for the extended model.

γ [(km×W)${}^{-1}$]	Q [W/(km×Hz)]	*L* [km]
1.25	$5.94\times 10^{-21}$	800

**Table 3 entropy-22-00607-t003:** Time grid spacings.

$\Delta_{1}$ [s]	$\Delta_{2}$ [s]	$\Delta_{3}$ [s]
$9.77\times 10^{-14}$	$1.95\times 10^{-13}$	$3.91\times 10^{-13}$

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
