# Peer review of "Path Integral Approach to Nondispersive Optical Fiber Communication Channel"

_entropy, 2020, doi:10.3390/e22060607_

Round 1

Reviewer 1 Report

The authors have investigated the channel estimation of optical fiber at larger SNR regime, based on the model of pre-sample and time-dependent model. The mutual information of the optical communication system using the non-dispersive optical fiber has been evaluated. The manuscript is interesting and useful for the design of optical fiber communication systems, and is acceptable to be published in Entropy, provided the following issue can be addressed

  1. Specify the full names of all abbreviations
  2. Add a table to summarize the parameters used in the model analysis and the numerical simulations.
  3. Improve the quality of Figure 1, Figure 2 and Figure 6.
  4. Detail the mutual information estimation procedure in numerical simulations.
  5. Discuss capacity and MI in the optical communication systems and the potential change in the model if the impact of fiber dispersion is considered

See. e.g.

  • D Semrau et al., Achievable information rates estimates in optically amplified transmission systems using nonlinearity compensation and probabilistic shaping, Optics Letters, 2017.
  • Liga et al., On the performance of multichannel digital backpropagation in high-capacity long-haul optical transmission,Optics Express, 2014.

Author Response

Dear Referee, we would like to thank you for observational reading and
constructive comments. Below we answer to your comments:

\subsection*{Comment 1}

{}{Specify the full names of all abbreviations}

\textbf{Response to comment 1}\\
At the Section Abbreviations we added missing abbreviations.

\subsection*{Comment 2}
{}{Add a table to summarize the parameters used in the model
analysis and the numerical simulations.}

\textbf{Response to comment 2}\\
For the reader convenience we added three tables on the page 18 and
21. In the first table we presented the parameters for the
per-sample channel which were used for the numerical calculation of
the lower bound. In the second table for the extended model we
presented the channel parameter which were used for the numerical
calculations. In the third one we presented the the grid spacings
for which the numerical calculations were performed.

\subsection*{Comment 3}
{}{Improve the quality of Figure 1, Figure 2 and Figure 6.}

\textbf{Response to comment 3}\\
We improved these figures.

\subsection*{Comment 4}
{}{Detail the mutual information estimation procedure in numerical
simulations.}

\textbf{Response to comment 4}\\

To clarify the numerical calculations of the lower bound for the
per-sample model we added the following sentence on the page 18:
``For the presented parameters we have calculated numerically the
lower bound $C_0$ using Eqs. (77) and (82).''

For the extended model (Section 3.2) we did not perform the
numerical simulations of the mutual information estimation
procedure. Instead of that, we calculated correlators of the output
signal (95)--(97) (in the first version of the manuscript) with the
accuracy ${\cal o}(1/\mathrm{SNR})$ and compared with the analytical
results. We demonstrated that the results coincides with the
necessary accuracy. This fact is demonstrated in Figs. 4 and 5.
Using analytical results for the correlators we built the
conditional PDF which reproduces all the correlators with necessary
accuracy. Then using this conditional PDF we found analytically the
mutual information (113). Since Eq.~ (113) is completely equivalent
of the Eq.~(80), we can use the method of the numerical calculation
developed for the per-sample channel (calculations of Eqs. (75) and
(80)).

\subsection*{Comment 5}
{}{Discuss capacity and MI in the optical communication systems and
the potential change in the model
if the impact of fiber dispersion is considered See. e.g.\\
D Semrau et al., Achievable information rates estimates in optically
amplified transmission systems using nonlinearity compensation and
probabilistic shaping, Optics Letters, 2017. \\ Liga et al., On the
performance of multichannel digital back propagation in
high-capacity long-haul optical transmission,Optics Express, 2014.}

\textbf{Response to comment 5}\\

In the present manuscript we concentrated only on the channels with
zero dispersion. Therefore, we did not discuss the channels with
nonzero dispersion in details. We only mentioned that these two
problems are substantially different. So, we added the References
noted by the Referee to the Introduction. Moreover, we stress the
difference of the two problems in the Conclusion adding the
following text: \\``The models considered in the present paper are
widely-spaced from the modern communication systems where the
coefficient of the second dispersion is not zero and the signal
detection procedure differs from considered above. The effects
related to the non-zero dispersion and properties of the receiver
can significantly change the results for the mutual information
obtained in our consideration. However, the methods described in the
present paper may be useful for the consideration of real
communication systems. ''

In conclusion we would like to thank the you again for very substantial reading and constructive comments that, we hope, will help us to improve the quality of the paper.

 Sincerely yours,  Ivan Terekhov, Alexey Reznichenko.

Reviewer 2 Report

In this paper, the authors present a method to estimate the mutual information and the channel capacity in a nondispersive optical fiber channel. The approach is interesting, and the paper is clear and well organized. Nevertheless, the results obtained are not impressive, indeed the constraints to the problem that they assume are quite substantial, namely to the channel model and to the input signals. There are in the literature other results where more realistic channel models and input signal models were assumed and more realistic lower bounds for the channel capacity were obtained. In my opinion, the interest of the paper resides more in the use of the path integral approach than in the relevance of the results.

The same authors have been publishing other papers based on the same method. In this paper, the authors perform a review of their previous work, but besides that, the authors should make clear in the introduction which is the novelty that this paper brings considering the previous papers published by the same authors.

The following technical aspects should also be made clearer in the paper:

1) The assumed detection process. It seems that it is assumed that the receiver has access to both the amplitude and phase of the signal, so some sort of coherent detection scheme is assumed. This should be made clear.

2) It should also be explained why the square operation in the photodetectors is not relevant.

3) The authors state that in the considered case “the bandwidth of the receiver coincides with the bandwidth of the noise”, which according to expression (15) is much larger than the signal bandwidth. This seems to be a gross approximation because to maximize the capacity, and therefore to minimize the noise, the bandwidth of the receiver should be close to the signal bandwidth, not noise bandwidth. This should be better discussed.

4) In order to obtain the capacity presented in expression (29), the authors assume that the phase does not transfer any information, but according to expression (16), it seems that it is assumed that the receiver is able to compensate for the nonlinear phase rotations. This should be better discussed.

5) In the major results presented, in section (3.2), the authors consider the Gaussian envelope, expression (99), but it is well known that the gaussian enveloped leads to inter-symbolic interference. How was that considered?

Author Response

Dear Referee, we would like to thank you for observational reading
and constructive comments. Below we answer to your comments:

\subsection*{Comment 0}
{}{The same authors have been publishing other papers based on the
same method. In this paper, the authors perform a review of their
previous work, but besides that, the authors should make clear in
the introduction which is the novelty that this paper brings
considering the previous papers published by the same authors.}

\textbf{Response to comment 0}\\

In the presented manuscript we obtained the lower bound of the
channel capacity for the per-sample model in the intermediate power
range. The Referee is right --- in the our manuscript we presented
the review of our previous papers. But in our previous papers we did
not find the lower bound but only the result for the capacity with
some accuracy, i.e., the asymptotic estimate for the capacity. This
was mentioned in the Introduction. In the second sentence of the
last paragraph of the Introduction we wrote: ``For the per-sample
model we find the lower bound of the channel capacity and compare
our results with ones obtained earlier.''

\subsection*{Comment 1}
{}{The assumed detection process. It seems that it is assumed that
the receiver has access to both the amplitude and phase of the
signal, so some sort of coherent detection scheme is assumed. This
should be made clear.}

\textbf{Response to comment 1}\\
Of course, it is clear from the detailed description of the
detection procedure in the subsection 2.2, the paragraph after
Eq.~(15) (numeration in the first version of the manuscript), that
our receiver measures both the phase and the amplitude.

\subsection*{Comment 2}
{}{It should also be explained why the square operation in the
photodetectors is not relevant.}

\textbf{Response to comment 2}\\

We did not state that the square operation in the photodetectors is
not relevant. We just stated that the we can measure the amplitude
and phase of the output signal. Moreover, we considered the specific
model of the detection procedure which includes the backward
propagation, see Eq.~ (16). To clarify this for a reader we add the
explanation after Eq.~(17) in the new version of the manuscript.

\subsection*{Comment 3}
{}{The authors state that in the considered case “the bandwidth of
the receiver coincides with the bandwidth of the noise”, which
according to expression (15) is much larger than the signal
bandwidth. This seems to be a gross approximation because to
maximize the capacity, and therefore to minimize the noise, the
bandwidth of the receiver should be close to the signal bandwidth,
not noise bandwidth. This should be better discussed.}

\textbf{Response to comment 3}\\

Yes, it is the case. Our aim was to demonstrate the path integral
approach is the useful method to consider the systems like that.
Using this approach the different types of the receivers can be
treated. We decide to choose the model receiver described in our
manuscript for clarity. Moreover, we have demonstrated that some
correlators depends on the noise power within the total bandwidth of
the noise ($W'$ in our notations).

\subsection*{Comment 4}
{}{In order to obtain the capacity presented in expression (29), the
authors assume that the phase does not transfer any information, but
according to expression (16), it seems that it is assumed that the
receiver is able to compensate for the nonlinear phase rotations.
This should be better discussed.}

\textbf{Response to comment 4}\\

Probably, it is misunderstanding. The result (29) was obtained by
the authors of Ref.~[15], where it had been assumed that the phase
of the signal did not carry the information. This statement is
correct for very large signal powers $P$, that was mentioned in
Refs.~[15] and [16]. In our manuscript we consider the intermediate
power range, see Eq.~ (31). For such signal powers the phase does
carry the information. It is discussed in the subsection 3.1.1 (the
text after Eq.~(30)) and in the subsection 3.1.3 (the text after
Eq.~ (68)).

\subsection*{Comment 5}
{}{In the major results presented, in section (3.2), the authors
consider the Gaussian envelope, expression (99), but it is well
known that the gaussian enveloped leads to inter-symbolic
interference. How was that considered?}

\textbf{Response to comment 5}\\
Yes, it is correct. But choosing the parameter of the envelope
($T_0$ and $T_1$ in our notations) we can achieve the negligibility
of the overlapping effects. To clarify this point we added the
paragraph (before the Conclusion) at the end of the Section 3. We
also added the sentence about this point in the subsection 2.2 after
Eq.~(11) in the previous version of the manuscript (after Eq.~(12)
in the new version).

In conclusion we would like to thank you again for very substantial reading
and constructive comments that, we hope, will help us to improve the quality of the paper.

Sincerely yours,

Ivan Terekhov, Alexey Reznichenko.

Round 2

Reviewer 1 Report

This paper can be accepted now.

Reviewer 2 Report

I believe that now the paper is in conditions to be accepted for publication.